# A Case for Object Compositionality in Deep Generative Models of Images

## Abstract

Deep generative models seek to recover the process with which the observed data was generated. They may be used to synthesize new samples or to subsequently extract representations. Successful approaches in the domain of images are driven by several core inductive biases. However, a bias to account for the compositional way in which humans structure a visual scene in terms of objects has frequently been overlooked. In this work we propose to structure the generator of a GAN to consider objects and their relations explicitly, and generate images by means of composition. This provides a way to efficiently learn a more accurate generative model of real-world images, and serves as an initial step towards learning corresponding object representations. We evaluate our approach on several multi-object image datasets, and find that the generator learns to identify and disentangle information corresponding to different objects at a representational level. A human study reveals that the resulting generative model is better at generating images that are more faithful to the reference distribution.

## 1 Introduction

Generative modelling approaches to representation learning seek to recover the process with which the observed data was generated. It is postulated that knowledge about the generative process exposes important factors of variation in the environment (captured in terms of latent variables) that may subsequently be obtained using an appropriate posterior inference procedure. Therefore, the *structure* of the generative model is critical in learning corresponding representations.

Deep generative models of images rely on the expressiveness of neural networks to learn the generative process directly from data (Goodfellow et al., 2014; Kingma & Welling, 2014; Van Oord et al., 2016). Their structure is determined by the *inductive bias* of the neural network, which steers it to organize its computation in a way that allows salient features to be recovered and ultimately captured in a representation (Dinh et al., 2017; Donahue et al., 2017; Dumoulin et al., 2017; Kingma & Welling, 2014). Recently, it has been shown that independent factors of variation, such as pose and lighting of human faces may be recovered in this way (Chen et al., 2016; Higgins et al., 2017).

A promising but under-explored inductive bias in deep generative models of images is *compositionality at the representational level of objects*, which accounts for the compositional nature of the visual world and our perception thereof (Battaglia et al., 2013; Spelke & Kinzler, 2007). It allows a generative model to describe a scene as a composition of objects (entities), thereby disentangling visual information in the scene that can be processed largely independent of one another. It provides a means to efficiently learn a more accurate generative model of real-world images, and by explicitly

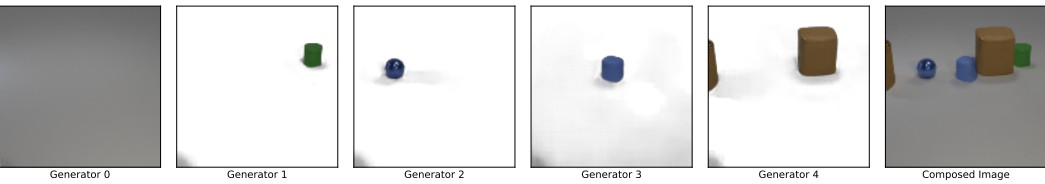

Generator 0   Generator 1   Generator 2   Generator 3   Generator 4   Composed Image

Figure 1: A scene (right) is generated as a composition of objects and background.

considering objects at a representational level, it serves as an important first step in recovering corresponding object representations.

In this work we investigate object compositionality for Generative Adversarial Networks (GANs; Goodfellow et al. (2014)), and present a general mechanism that allows one to incorporate corresponding structure in the generator. Starting from strong independence assumptions about the objects in images, we propose two extensions that provide a means to incorporate dependencies among objects and background. In order to efficiently represent and process multiple objects with neural networks, we must account for the binding problem that arises when superimposing multiple distributed representations (Hinton et al., 1984). Following prior work, we consider different representational slots for each object (Greff et al., 2017; Nash et al., 2017), and a relational mechanism that preserves this separation accordingly (Zambaldi et al., 2018).

We evaluate our approach[1] on several multi-object image datasets, including three variations of Multi-MNIST, a multi-object variation of CIFAR10, and CLEVR. In particular the latter two mark a significant improvement in terms of complexity, compared to datasets that have been considered in prior work on unconditional multi-object image generation and multi-object representation learning.

In our experiments we find that our generative model learns about the individual objects and the background of a scene, without prior access to this information. By disentangling this information at a representational level, it generates novel scenes efficiently through composing individual objects and background, as can be seen in Figure 1. As a quantitative experiment we compare to a strong baseline of popular GANs (Wasserstein and Non-saturating) with recent state-of-the-art techniques (Spectral Normalization, Gradient Penalty) optimized over multiple runs. A human study reveals that the proposed generative model outperforms this baseline in generating better images that are more faithful to the reference distribution.

## 2 GENERATIVE ADVERSARIAL NETWORKS

Generative Adversarial Networks (GANs; Goodfellow et al. (2014)) are a powerful class of generative models that learn a stochastic procedure to generate samples from a distribution $P(X)$. Traditionally GANs consist of two deterministic functions: a generator $G(z)$ and a discriminator (or critic) $D(x)$. The goal is to find a generator that accurately transforms samples from a prior distribution $z \sim P(Z)$ to match samples from the target distribution $x \sim P(X)$. This can be done by using the discriminator to implement a suitable objective for the generator, in which it should behave *adversarial* with respect to the goal of the discriminator in determining whether samples $x$ were sampled from $P(X)$ or $G(P(Z))$ respectively. These objectives can be summarized as a *minimax* game with the following value function:

$$\min_G \max_D V(D, G) = \mathbb{E}_{x \sim P(X)} \left[ \log D(x) \right] + \mathbb{E}_{z \sim P(Z)} \left[ \log(1 - D(G(z))) \right]. \quad (1)$$

When the generator and the discriminator are implemented with neural networks, optimization may proceed through alternating (stochastic) gradient descent updates of their parameters with respect to (1). However, in practice this procedure might be unstable and the minimax formulation is known to be hard to optimize. Many alternative formulations have been proposed and we refer the reader to Lucic et al. (2017) and Kurach et al. (2018) for a comparison.

Following the recommendations of Kurach et al. (2018) we consider two practical reformulations of (1) in this paper: Non-Saturating GAN (NS-GAN; Goodfellow et al. (2014)), in which the generator maximizes the probability of generated samples being real, and Wasserstein GAN (WGAN; Arjovsky et al. (2017)) in which the discriminator minimizes the Wasserstein distance between $G(P(Z))$ and $P(X)$. For both formulations we explore two additional techniques that have proven to work best on a variety of datasets and architectures: the gradient penalty from Gulrajani et al. (2017) to regularize the discriminator, and spectral normalization (Miyato et al., 2018) to normalize its gradient flow.

---

[1]Code is already available online. link is not added to preserve anonymity.

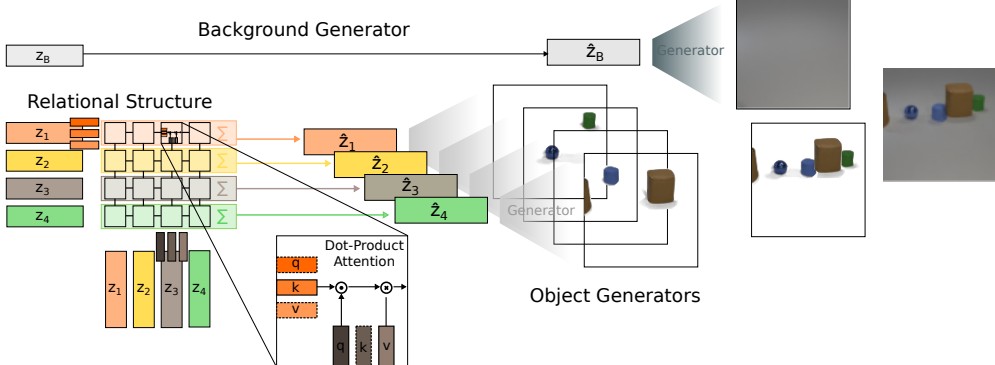

Figure 2: We propose to structure the generator of a GAN to generate images as compositions of individual objects and background. In this case it consists of $K = 4$ *object generators* (shared weights) that each generate an image from a separate latent vector $\hat{z}_i$. These are obtained by having each $z_i \sim P(Z)$ participate in a *relational stage*, which allows each representation to be updated as a function of all others. Alternatively $\hat{z}_i = z_i$ if no relations are to be modelled. On the top, a *background generator* (unique weights) generates a background image from a separate latent vector $z_b \sim P(Z_b)$, which optionally participates in the relational stage. The whole system is trained end-to-end as in the standard GAN framework, and the final image is obtained by composing (in this case using alpha compositing) the outputs of all generators.

## 3 INCORPORATING STRUCTURE

In order to formulate the *structure* required to achieve object compositionality in neural networks we primarily focus on the corresponding type of generalization behavior that we are interested in.[2] It is concerned with independently varying the different visual primitives (objects) that an image is composed of, requiring these to be identified at a representational level and described in a common format. We account for the binding problem (Hinton et al., 1984; Milner, 1974; Von Der Malsburg, 1994) that may arise in combining these object representations to arrive at a final image.

In the following subsections we initially present structure that assumes strict object independence (Section 3.1), to then relax this assumption by incorporating relational structure (Section 3.2), and finally allow for the possibility of unstructured background and occlusion (Section 3.3).

### 3.1 STRICT INDEPENDENCE

If we assume that images in $P(X)$ are composed of objects that are strictly independent of one another then (without loss of generality) we may structure our latent variables accordingly. For images having $K$ objects, we consider $K$ i.i.d. vector-valued random variables $Z_i$ that each describe an object at a representational level. $K$ copies of a deterministic generator $G(z)$ transform samples from each $Z_i$ into images, such that their superposition results in the corresponding scene:

$$G_{multi}([z_1, \cdots, z_K]) = \sum_{1}^{K} G(z_i) \ , \ z_i \sim P(Z) \tag{2}$$

When each copy of $G$ generates an image of a single object, the resulting generative model efficiently describes images in $P(X)$ in a compositional manner. Each object in (2) is described in terms of the same features (i.e. the $Z_i$'s are i.i.d) and the weights among the generators are shared, such that any acquired knowledge in generating a specific object is transferred across all others. Hence, rather than having to learn about all combinations of objects (including their individual variations) that may appear in an image, it suffices to learn about the different variations of each individual object instead.

---

[2]Following Battaglia et al. (2018) we define structure as "the product of composing a set of known building blocks". *Structured representations* then capture this composition, and *structured computations* operate over the elements and their composition as a whole.

Notice that the generators in (2) cannot communicate, which prevents degenerate solutions from being learned. This comes at a cost in that relations among the objects cannot be modelled in this way. An additional concern is the sum in (2), which assumes that images only consist of objects, and that their values can be summed in pixel-space. We will address these concerns in the following, using the superposition of generators as a backbone for object compositionality in our approach.

## 3.2 RELATIONAL STRUCTURE

In the real world objects are not strictly independent of one another. Certain objects may only occur in the presence of others, or affect their visual appearance in subtle ways (eg. shadows, lighting). In order to incorporate relationships of this kind we introduce a *relational stage*, in which the representation of an object is *updated* as a function of all others, before each generator proceeds to generate its image.

Following Zambaldi et al. (2018) we consider one or more "attention blocks" to compute interactions among the object representations. At its core is Multi-Head Dot-Product Attention (MHDPA; Vaswani et al. (2017)) that performs non-local computation (Wang et al., 2018) or message-passing (Gilmer et al., 2017) when one associates each object representation with a node in a graph. When specific design choices are made, computation of this kind provides an efficient means to learn about relations between objects and update their representations accordingly (Battaglia et al., 2018).

A single head of an attention block updates $z_i$ in the following way:

$$\boldsymbol{q}_i, \boldsymbol{k}_i, \boldsymbol{v}_i = \mathrm{MLP}^{(\cdot)}(\boldsymbol{z}_i) \quad \boldsymbol{A} = \underbrace{softmax\big(\frac{\boldsymbol{Q}\boldsymbol{K}^T}{\sqrt{d}}\big)}_{\text{attention weights}} \boldsymbol{V} \quad \hat{\boldsymbol{z}}_i = \mathrm{MLP}^{up}(\boldsymbol{a}_i) + \boldsymbol{z}_i \quad (3)$$

where $d = dim(\boldsymbol{v}_i)$ and each MLP corresponds to a multi-layer perceptron. First, a query vector $\boldsymbol{q}_i$, a value vector $\boldsymbol{v}_i$, and a key vector $\boldsymbol{k}_i$ is computed for each $\boldsymbol{z}_i$. Next, the interaction of an object $i$ with all other objects (including itself) is computed as a weighted sum of their value vectors. Weights are determined by computing dot-products between $\boldsymbol{q}_i$ and all key vectors, followed by softmax normalization. Finally, the resulting update vector $\boldsymbol{a}_i$ is projected back to the original size of $\boldsymbol{z}_i$ using MLP$^{up}$ before being added.

Additional heads (modelling different interactions) use different parameters for each MLP in (3). In this case their outputs are combined with another MLP to arrive at a final $\boldsymbol{z}_i$. Complex relationships among objects can be modelled by using multiple attention blocks to *iteratively* update $\boldsymbol{z}_i$. A detailed overview of these computations can be found in Appendix B, and an overview in Figure 2.

## 3.3 INCORPORATING BACKGROUND AND ALPHA COMPOSITING

Up until this point we have assumed that an image is entirely composed of objects, which may be prohibitive for complex visual scenes. For example, certain objects that only appear in the "background" may not occur frequently enough, nor have a regular visual appearance that allows a model to consider them as such. One could reason that certain visual primitives (those that can be varied independently and re-composed accordingly) will be discovered from the observed data, whereas all other remaining visual information is captured as *background* by another component. However, these are conflicting assumptions as the latent representations $\boldsymbol{z}_i$ (and corresponding generator) now need to describe objects that assume a regular visual appearance, as well as background that is not regular in its visual appearance at all. Therefore, we consider an additional generator (see Figure 2) having its own set of weights to generate the background from a separate vector of latent variables $\boldsymbol{z}_b \sim P(Z_b)$. We consider two different variations of this addition, one in which $z_b$ participates in the relational stage, and one in which it does not.

A remaining challenge is in *combining* objects with background and occlusion. A straightforward adaptation of the sum in (2) to incorporate pixel-level weights would require the background generator to assign a weight of zero to all pixel locations where objects appear, thereby increasing the complexity of generating the background exponentially. Instead, we require the object generators to generate an additional alpha channel for each pixel, and use alpha compositing to combine the outputs of the different generators and background through repeated application of:

$$\boldsymbol{x}_{new} = \frac{\boldsymbol{x}_i \boldsymbol{\alpha}_i + \boldsymbol{x}_j \boldsymbol{\alpha}_j (1 - \boldsymbol{\alpha}_i)}{\boldsymbol{\alpha}_{new}} \qquad \boldsymbol{\alpha}_{new} = \boldsymbol{\alpha}_i + \boldsymbol{\alpha}_j (1 - \boldsymbol{\alpha}_i) \qquad (4)$$

## 4 RELATED WORK

Inductive biases aimed at object compositionality have been previously explored, both in the context of generative models and multi-object representation learning. One line of work models an image as a spatial mixture of image patches, utilizing *multiple copies* of the same function to arrive at a compositional solution. Different implementations consider RBMs (Le Roux et al., 2011), VAEs (Nash et al., 2017), or (recurrent) auto-encoders inspired by EM-like inference procedures (Greff et al., 2016; 2017) to generate these patches. They consider objects at a representational level and recent work has shown a means to efficiently model interactions between them (van Steenkiste et al., 2018). However, neither of these approaches are capable of modelling complex visual scenes that incorporate unstructured background as well as interactions among objects.

A conceptually different line of work relies on recurrent neural networks to iteratively model multiple objects in an image, one at a time. Gregor et al. (2015) proposes to use attention to arrive at this solution, whereas Eslami et al. (2016) considers objects explicitly. This approach has also been explored in the context of GANs. Im et al. (2016) generates images iteratively by accumulating outputs of a recurrent generator, and Kwak & Zhang (2016) propose to combine these outputs using alpha compositing. Yang et al. (2017) extends this approach further, by considering a separate generator for the background, using spatial transformations to integrate a foreground image. They briefly explore multi-object image generation on a dataset consisting of two non-overlapping MNIST digits, yet their approach requires prior knowledge about the size of the objects, and the number of objects to generate. This is information typically unavailable in the real world and not required for our method. A more general concern is the difficulty in modelling relations among objects when they are generated one at a time. Information about the objects must be stored and updated in the memory of the RNN, which is ill-suited for this task without incorporating corresponding relational structure (Santoro et al., 2018). It prevents relations from being learned efficiently, and requires the RNN to commit to a plan in its first step, without the possibility to revisit this decision.

Recent work in GANs is increasingly focusing on incorporating (domain-specific) architectural structure in the generator to generate realistic images. Lin et al. (2018) considers a Spatial Transformer Network as a generator, and proposes an iterative scheme to remove or add objects to a scene. Johnson et al. (2018) propose image generation by conditioning on explicit scene graphs to overcome the limitations of standard GANs in generating scenes composed of multiple objects that require relations to be taken into account. Xu et al. (2018) propose a similar approach but condition on a stochastic and-or graph instead. Azadi et al. (2018) considers a framework to generate images composed of two objects, conditioned on images of each single object. In our approach we make use of an implicit graph structure (as implemented by our relational mechanism) to model relations among objects, and do not rely on prior information about individual objects (in the form of conditioning).

## 5 EXPERIMENTS

We test different aspects of the proposed structure on several multi-object datasets. We are particularly interested in verifying that images are generated as compositions of objects and that the relational and background structure is properly utilized. To that extend, we study how the incorporated structure affects the quality and the content of generated images.

**Datasets** We consider five multi-object datasets.[3] The first three are different variations of *Multi-MNIST (MM)*, in which each image consists of three MNIST digits that were rescaled and drawn randomly onto a $64 \times 64$ canvas. In *Independent MM*, digits are chosen randomly and there is no relation among them. The *Triplet* variation requires that all digits in an image are of the same type, requiring relations among the digits to be considered during the generative process. Similarly *RGB Occluded MM* requires that each image consist of exactly one red, green, and blue digit. The fourth dataset (*CIFAR10 + MM*) is a variation of CIFAR10 (Krizhevsky & Hinton, 2009) in which the

---

[3]Datasets are already available online, link is not added to preserve anonymity.

digits from *RGB Occluded MM* are drawn onto a randomly choosen (resized) CIFAR10 image. Our final dataset is CLEVR (Johnson et al., 2017), which we downsample to $160 \times 240$ followed by center-cropping to obtain $128 \times 128$ images. Samples from each dataset can be seen in in Appendix C.

**Evaluation**    A popular evaluation metric in comparing generated images by GANs is the Fréchet Inception Distance (FID; Heusel et al. (2017)). It computes the distance between two empirical distributions of images (one generated, and a reference set) as the Fréchet (Wasserstein-2) distance between two corresponding multivariate Gaussian distributions that were estimated from the Inception-features computed for each image. Although pior work found that FID correlates well with perceived human quality of images on standard image datasets Heusel et al. (2017); Lucic et al. (2017), we find that FID is of limited use when considering image datasets in which the dominant visual aspects are determined by *multiple* objects. Our results in Section 5.2 suggest that FID can not be used to verify whether image distributions adhere to certain properties, such as the number of objects. We hypothesize that this inability is inherent to the Inception embedding having been trained only for single object classification.

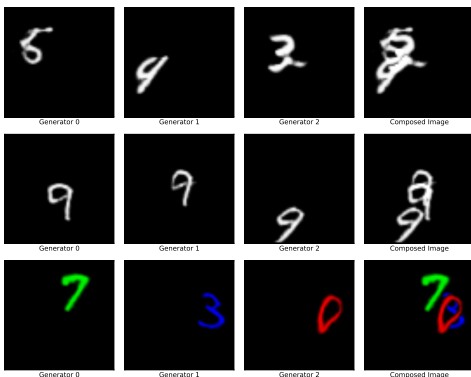

Figure 3: Generated samples by *3-GAN* on Multi-MNIST: *Independent* (top), *Triplet* (middle), and *RGB Occluded* (bottom). The three columns on the left show the output of each object generator, and the right column the composed image.

To compensate for this we conduct two different studies among humans, 1) to compare images generated by our models to a baseline, and 2) to answer questions about the content of generated images. The latter allows us to verify whether generated images are probable samples from our image distribution, eg. by verifying that they have the correct number of objects. As conducting human evaluation of this kind is not feasible for large-scale hyper-parameter search we will continue to rely on FID to select the "best" models during hyper-parameter selection. Details of these human studies can be found in Appendix B.

**Set-up**    Each model is optimized with ADAM (Kingma & Ba, 2015) using a learning rate of $10^{-4}$, and batch size 64 for 1M steps. We compute the FID (using 10K samples) every 20K steps, and select the best set of parameters accordingly. On each dataset, we compare GANs that incorporate our proposed structure to a strong baseline that does not. In both cases we conduct extensive grid searches covering on the order of 40-50 hyperparameter configurations for each dataset, using ranges that were previously found good for GAN (Lucic et al., 2017; Kurach et al., 2018). Each configuration is ran with 5 different seeds to be able to estimate its variance. An overview of each hyper-parameter search can be found in Appendix B, and samples of our best models in Appendix C.

**Composing**    On *Independent MM* and *Triplet MM* we sum the outputs of the object generators as in (2), followed by clipping. On all other datasets we use alpha compositing (4) with a fixed order. In this case the object generators output an additional alpha channel, except for *RGB Occluded MM* in which we obtain alpha values by thresholding the output of each object generator for simplicity.

**Notation**    In reporting our results we will break down the results obtained when incorporating structure in GAN across the different structural parts. In particular we will denote *k-GAN* to describe a generator consisting of $K = k$ components, *k-GAN rel.* if it incorporates relational structure and *k-GAN ind.* if it does not. Additionally we will append "*bg.*" when the model includes a separate background generator. Since any variation incorporates multiple components, we will use *k-GAN* to refer to GANs that incorporate any of the proposed structure as a collective. We will use *GAN* to refer to the collection of GANs with different hyperparameters in our baseline.

## 5.1    QUALITATIVE ANALYSIS

**Utilizing Structure**    In analyzing the output of each generator for *k-GAN*, we consistently find that the final image is generated as a composition of images consisting of individual objects and

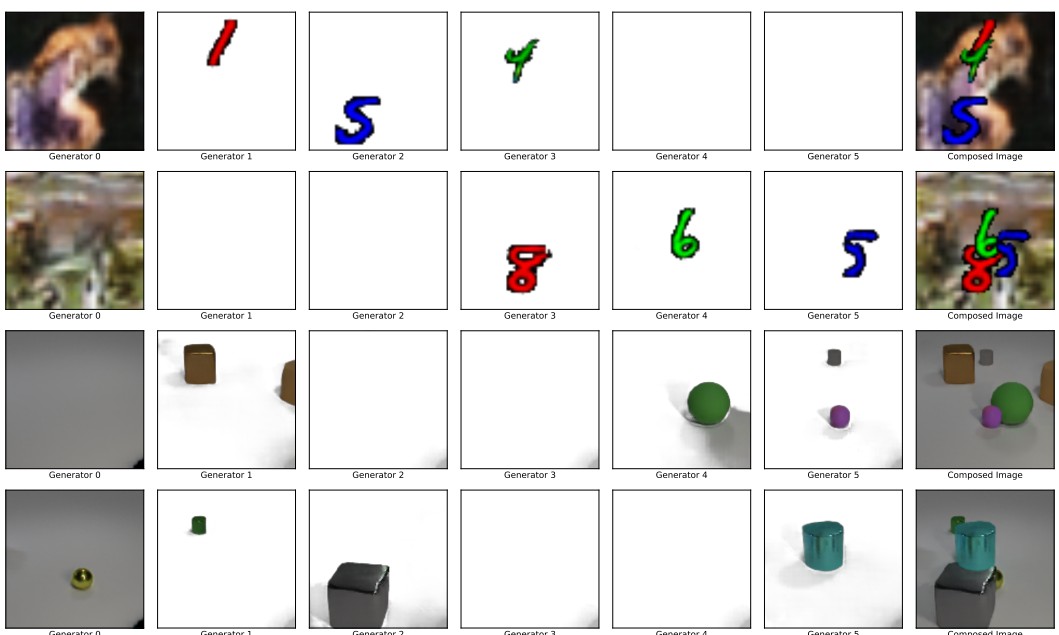

Figure 4: Generated samples by *5-GAN rel. bg.* on *CIFAR10 + MM* (top two), and CLEVR (bottom two). The left column corresponds to the output of the background generator. The first five columns are the outputs of each object generator, and the right column the composed image. Images are displayed as RGBA, with white denoting an alpha value of zero.

background. Hence, in the process of learning to generate images, *k-GAN* learns about what are individual objects, and what is background, without relying on prior knowledge or conditioning. By *disentangling* this information at the representational level, it opens the possibility to recover corresponding object representations. Examples for each dataset, can be seen in Figure 3 and Figure 4.

In the case of CLEVR, in which images may have a greater number of objects than the number of components $K$ that we used during training, we find that the generator continues to learn a factored solution. Visual primitives are now made up of multiple objects, examples of which can be seen at the bottom rows in Figure 4. A similar tendency was also found when analyzing generated images by *k-GAN ind.* when k > 3 on Multi-MNIST. The generator decodes part of its latent space as "no digit" as an attempt at generating the correct number of digits.

From the generated samples in Appendix C we observe that relations among the objects are correctly captured in most cases. In analyzing the background generator we find that it sometimes generates a single object together with the background. It rarely generates more than one object, confirming that although it is capable, it is indeed more efficient to generate images as compositions of objects.

**Latent Traversal**   We explore the degree to which the relational structure affects our initial independence assumption about objects. If it were to cause the latent representations to be fully dependent on one another then our approach would no longer be compositional in the strict sense. Note that although we have a clear intuition in how this mechanism should work, there is no corresponding constraint in the architecture. We conduct an experiment in which we traverse the latent space of a single latent vector in *k-GAN rel.*, by adding a random vector to the original sample with fixed increments and generating an image from the resulting latent vectors. Several examples can be seen in Figure 5b. In the first row it can be seen that as we traverse the latent space of a single component the blue digit 9 takes on the shape of a 3, whereas the visual presentation of the others remain unaffected. Similarly in the second and third row the green digits are transformed, while other digits remain fixed. Hence, by disentangling objects at a representational level the underlying representation is more robust to common variations in image space.

We observe this behavior for the majority of the generated samples, confirming to a large degree our own intuition of how the relational mechanism should be utilized. When we conduct the same

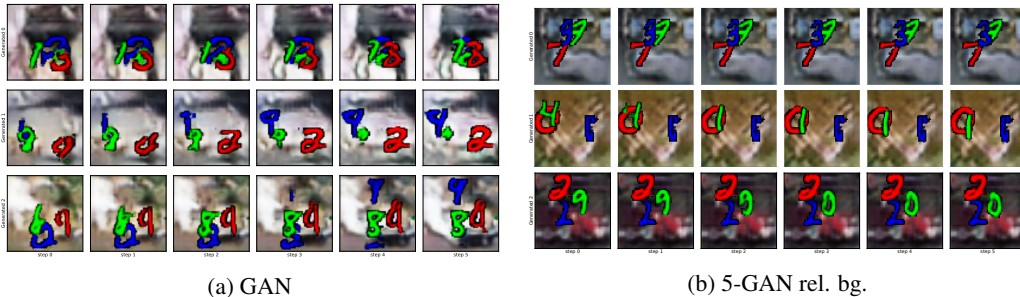

(a) GAN     (b) 5-GAN rel. bg.

Figure 5: Three generated images by a) *GAN* and b) *5-GAN rel. bg.*, when traversing the latent space of a single (object) generator at different increments. On the right it can be seen that in each case only a single digit is transformed, whereas the visual presentation of the others remains unaffected. In the case of GAN (left) the entire scene changes.

latent traversal on the latent space of *GAN* for which the information encoding different objects is entangled, it results in a completely different scene (see Figure 5a).

## 5.2 QUANTITATIVE ANALYSIS

**FID**   We train *k-GAN* and *GAN* on each dataset, and compare the FID of the models with the lowest average FID across seeds. On all datasets but CLEVR we find that *k-GAN* compares favorably to our baseline, although typically by a small margin. A break-down of the FID achieved by different variations of *k-GAN* reveals several interesting observations (Figure 9). In particular, it can be observed that the lowest FID on *Independent MM* is obtained by *4-GAN* without relational structure. This is surprising as each component is strictly independent and therefore *4-GAN ind.* is unable to consistently generate 3 digits. Indeed, if we take a look at the generated samples in Figure 11, then we frequently observe that this is the case. It suggests that FID is unable to account for these properties of the generated images, and renders the small FID differences that we observed inconclusive. Figure 9 does reveal some large FID differences across the different variations of *k-GAN* on *Triplet MM*, and *RGB Occluded MM*. It can be observed that the lack of a relational mechanism on these datasets is prohibitive (as one would expect), resulting in poor FID for *k-GAN ind.* Simultaneously it confirms that the relational mechanism is properly utilized when relations are present.

**Human evaluation**   We asked humans to compare the images generated by *k-GAN rel.* (k=3,4,5) to our baseline on *RGB Occluded MM*, *CIFAR10 + MM* and *CLEVR*, using the configuration with a background generator for the last two datasets. For each model we select the 10 best hyper-parameter configurations (lowest FID), from which we each generate 100 images. We asked up to three raters for each image and report the majority vote or "Equal" if no decision can be reached.

Figure 6a reports the results when asking human raters to compare the visual quality of the generated images by *k-GAN* to those by *GAN*. It can be seen that *k-GAN* compares favorably across all datasets, and in particular on *RGB Occluded MM* and *CIFAR10 + MM* we observe large differences. We find that *k-GAN* performs better even when $k > 3$, which can be attributed to the relational mechanism, allowing all components to agree on the correct number of digits.

In a second study we asked humans to report specific properties of the generated images (number of objects, number of digits, etc.), a complete list of which can be found in Appendix B. Here our goal was to asses if the generated images by *k-GAN* are more faithful to the reference distribution. The results on *RGB Occluded MM* are summarized in Figure 6b. It can be seen that *k-GAN* more frequently generates images that have the correct number of objects, number of digits, and that satisfy all properties simultaneously (color, digit count, shapes). The difference between the correct number of digits and correct number of objects suggests that the generated objects are often not recognizable as digits. This does not appear to be the case from the generated samples in Appendix C, suggesting that the raters may not have been familiar enough with the variety of MNIST digits.

On *CIFAR10 + MM* (Figure 7a) it appears that *GAN* is able to accurately generate the correct number of objects, although the addition of background makes it difficult to provide a comparison in this case.

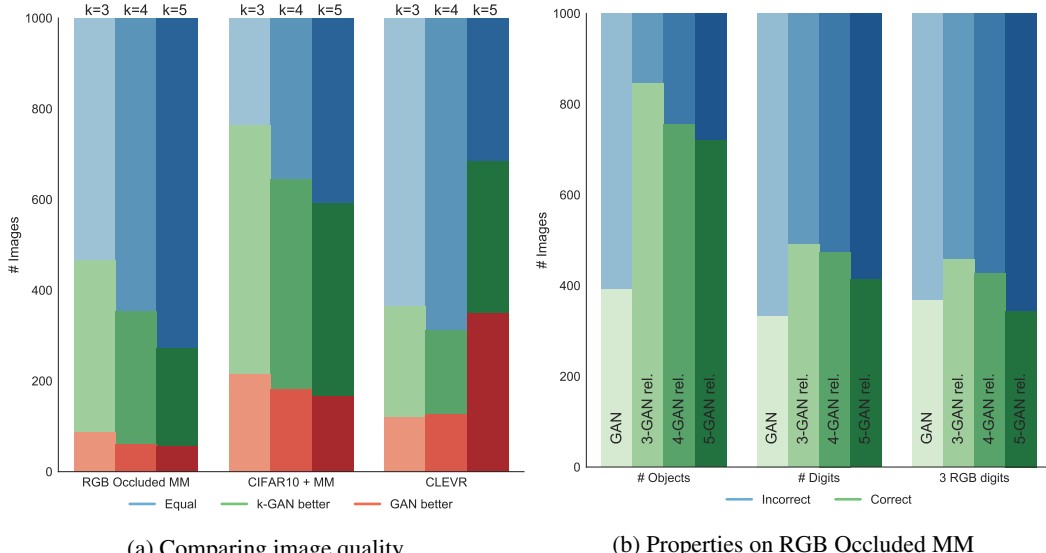

(a) Comparing image quality        (b) Properties on RGB Occluded MM

Figure 6: Results of human evaluation a) comparing the quality of the generated images by *k-GAN* (k=3,4,5) to *GAN* b) Properties of generated images by *k-GAN* (k=3,4,5) and *GAN* on *RGB Occluded MM*. It can be seen that *k-GAN* generates better images (a) that are more faithful to the reference distribution (b).

On the other hand if we look at the number of digits, then we find that *k-GAN* outperforms *GAN* by the same margin, as one would expect compared to the results in Figure 6b.

In comparing the generated images by *k-GAN* and *GAN* on CLEVR we noticed that the former generated more crowded scenes (containing multiple large objects in the center), and more frequently generated objects with distorted shapes or mixed colors. On the other hand we found cases in which *k-GAN* generated scenes containing "flying" objects, a by-product of the fixed order in which we apply (4). We asked humans to score images based on these properties, which confirmed these observations (see Figure 7b), although some differences are small.

## 6 DISCUSSION

The experimental results confirm that the proposed structure is beneficial in generating images of multiple objects, and is utilized according to our own intuitions. In order to benefit maximally from this structure it is desirable to be able to accurately estimate the (minimum) number of objects in the environment in advance. This task is ill-posed as it relies on a precise definition of "object" that is generally not available. In our experiments on CLEVR we encounter a similar situation in which the number of components does not suffice the potentially large number of objects in the environment. Here we find that it does not render the proposed structure useless, but instead each component considers "primitives" that correspond to multiple objects.

One concern is in being able to accurately determine foreground, and background when combining the outputs of the object generators using alpha compositing. On CLEVR we observe cases in which objects appear to be flying, which is the result of being unable to route the information content of a "foreground" object to the corresponding "foreground" generator as induced by the fixed order in which images are composed. Although in principle the relational mechanism may account for this distinction, a more explicit mechanism may be preferred (Mena et al., 2018).

We found that the pre-trained Inception embedding is not conclusive in reasoning about the validity of multi-object datasets. Similarly, the discriminator may have difficulties in accurately judging images from real / fake without additional structure. Ideally we would have a discriminator evaluate the correctness of each object individually, as well as the image as a whole. The use of a patch discriminator (Isola et al., 2017), together with the alpha channel of each object generator to provide a segmentation, may serve a starting point in pursuing this direction.

## 7    CONCLUSION

We have argued for the importance of compositionality at the representational level of objects in deep generative models of images, and demonstrated how corresponding structure may be incorporated in the generator of a GAN. On a benchmark of multi-object datasets we have shown that the proposed generative model learns about individual objects and background in the process of synthesizing samples. A human study revealed that this leads to a better generative model of images. We are hopeful that in disentangling information corresponding to different objects at a representational level these may ultimately be recovered. Hence, we believe that this work is an important contribution towards learning object representations of complex real-world images without any supervision.

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

## A EXPERIMENT RESULTS

### A.1 HUMAN STUDY PROPERTIES

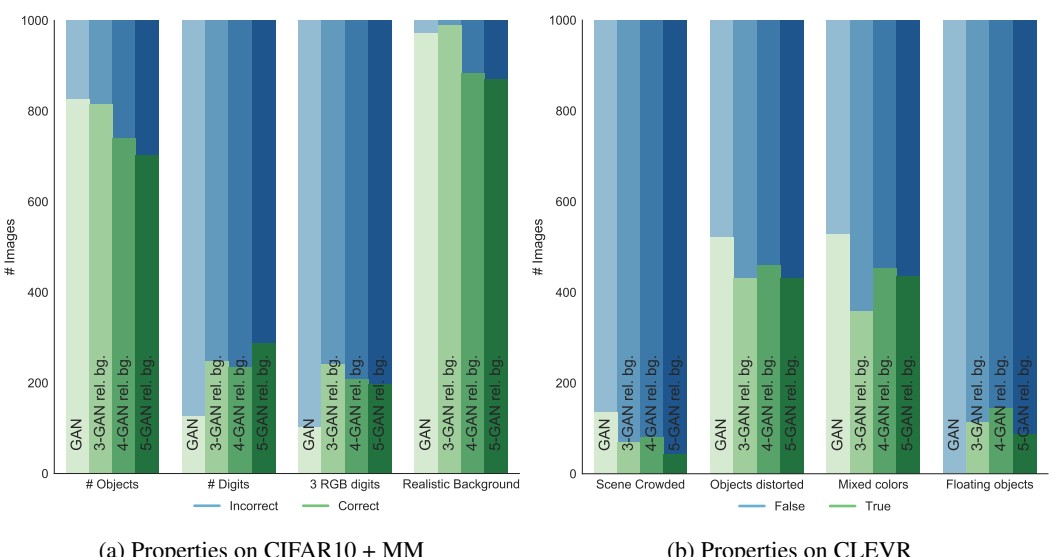

(a) Properties on CIFAR10 + MM

(b) Properties on CLEVR

Figure 7: Results of human evaluation. Properties of generated images by *k-GAN* (k=3,4,5) and *GAN* on *CIFAR10 + MM* (a) and CLEVR (b). Note that on CLEVR all evaluated properties are undesirable, and thus a larger number of "False" responses is better.

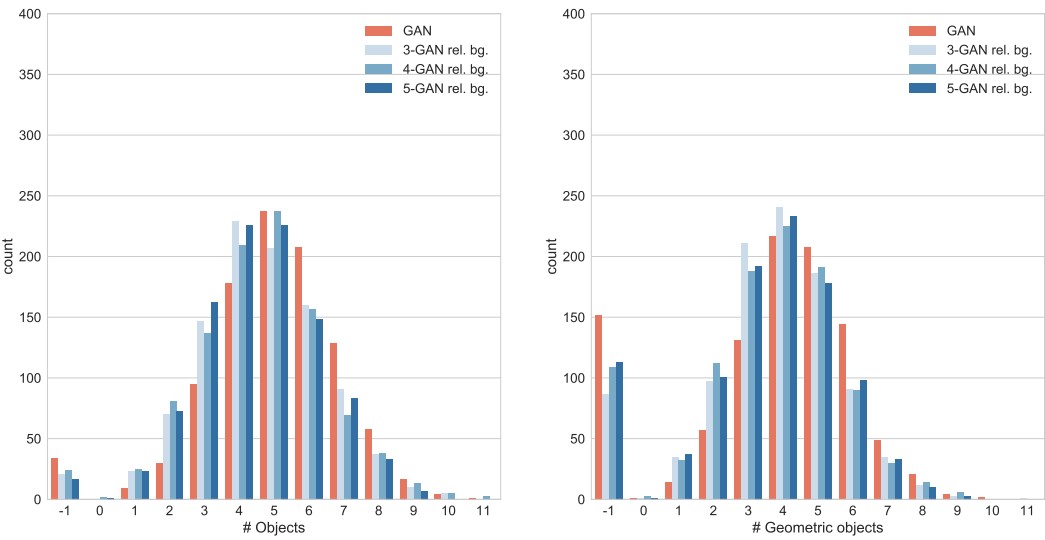

Figure 8: Results of human evaluation. Number of (geometric) objects in generated images by *k-GAN* (k=3,4,5) and *GAN* on CLEVR. A value of -1 implies a majority vote could not be reached.

## A.2 FID STUDY

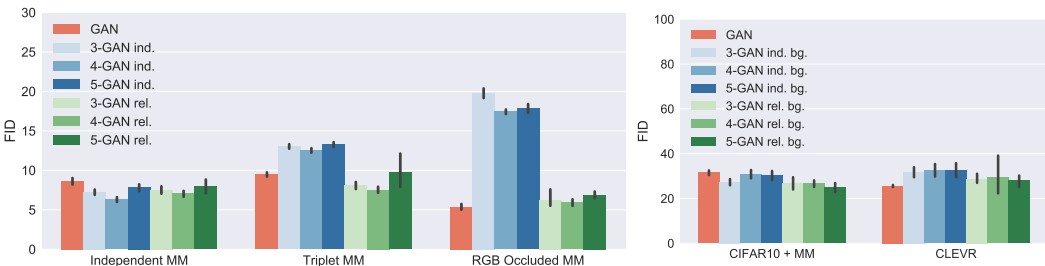

Figure 9: The best FID obtained by *GAN* and *k-GAN* following our grid search. The best configurations were chosen based on the smallest average FID (across 5 seeds). Standard deviations across seeds are illustrated with error bars.

### A.2.1 BEST CONFIGURATIONS

Table 1 reports the best hyper-parameter configuration for each model that were obtained following our grid search. Configurations were chosen based on the smallest average FID (across 5 seeds) as reported in Figure 9. Each block corresponds to a dataset (from top to bottom: *Independent MM*, *Triplet MM*, *RGB Occluded MM*, *CIFAR10 + MM*, *CLEVR*)

| model | gan type | norm. | penalty | blocks | heads | share | bg. int. | $\beta_1$ | $\beta_2$ | $\lambda$ |
|---|---|---|---|---|---|---|---|---|---|---|
| GAN | NS-GAN | spec. | none | x | x | x | x | 0.5 | 0.999 | 10 |
| 3-GAN ind. | NS-GAN | spec. | WGAN | x | x | x | x | 0.9 | 0.999 | 1 |
| 4-GAN ind. | NS-GAN | spec. | WGAN | x | x | x | x | 0.9 | 0.999 | 1 |
| 5-GAN ind. | NS-GAN | spec. | WGAN | x | x | x | x | 0.9 | 0.999 | 1 |
| 3-GAN rel. | NS-GAN | spec. | WGAN | 1 | 1 | no | x | 0.9 | 0.999 | 1 |
| 4-GAN rel. | NS-GAN | spec. | WGAN | 1 | 1 | no | x | 0.9 | 0.999 | 1 |
| 5-GAN rel. | NS-GAN | spec. | WGAN | 2 | 1 | no | x | 0.9 | 0.999 | 1 |
| GAN | NS-GAN | spec. | none | x | x | x | x | 0.5 | 0.999 | 1 |
| 3-GAN ind. | NS-GAN | spec. | WGAN | x | x | x | x | 0.9 | 0.999 | 1 |
| 4-GAN ind. | NS-GAN | spec. | WGAN | x | x | x | x | 0.9 | 0.999 | 1 |
| 5-GAN ind. | NS-GAN | spec. | WGAN | x | x | x | x | 0.9 | 0.999 | 1 |
| 3-GAN rel. | NS-GAN | spec. | WGAN | 1 | 1 | no | x | 0.9 | 0.999 | 1 |
| 4-GAN rel. | NS-GAN | spec. | WGAN | 1 | 2 | no | x | 0.9 | 0.999 | 1 |
| 5-GAN rel. | NS-GAN | spec. | WGAN | 2 | 1 | no | x | 0.9 | 0.999 | 1 |
| GAN | NS-GAN | spec. | none | x | x | x | x | 0.5 | 0.999 | 1 |
| 3-GAN ind. | NS-GAN | spec. | WGAN | x | x | x | x | 0.9 | 0.999 | 1 |
| 4-GAN ind. | NS-GAN | spec. | WGAN | x | x | x | x | 0.9 | 0.999 | 1 |
| 5-GAN ind. | NS-GAN | spec. | WGAN | x | x | x | x | 0.9 | 0.999 | 1 |
| 3-GAN rel. | NS-GAN | none | WGAN | 2 | 2 | yes | x | 0.9 | 0.999 | 1 |
| 4-GAN rel. | NS-GAN | none | WGAN | 2 | 2 | no | x | 0.9 | 0.999 | 1 |
| 5-GAN rel. | NS-GAN | none | WGAN | 2 | 2 | yes | x | 0.9 | 0.999 | 1 |
| GAN | NS-GAN | none | WGAN | x | x | x | x | 0.9 | 0.999 | 1 |
| 3-GAN ind. bg. | NS-GAN | none | WGAN | x | x | x | x | 0.9 | 0.999 | 1 |
| 4-GAN ind. bg. | NS-GAN | none | WGAN | x | x | x | x | 0.9 | 0.999 | 1 |
| 5-GAN ind. bg. | NS-GAN | none | WGAN | x | x | x | x | 0.9 | 0.999 | 1 |
| 3-GAN rel. bg. | NS-GAN | none | WGAN | 2 | 1 | yes | yes | 0.9 | 0.999 | 1 |
| 4-GAN rel. bg. | NS-GAN | none | WGAN | 2 | 1 | yes | yes | 0.9 | 0.999 | 1 |
| 5-GAN rel. bg. | NS-GAN | none | WGAN | 2 | 2 | yes | no | 0.9 | 0.999 | 1 |
| GAN | WGAN | none | WGAN | x | x | x | x | 0.9 | 0.999 | 1 |
| 3-GAN ind. bg. | NS-GAN | none | WGAN | x | x | x | x | 0.9 | 0.999 | 1 |
| 4-GAN ind. bg. | NS-GAN | none | WGAN | x | x | x | x | 0.9 | 0.999 | 1 |
| 5-GAN ind. bg. | NS-GAN | none | WGAN | x | x | x | x | 0.9 | 0.999 | 1 |
| 3-GAN rel. bg. | NS-GAN | none | WGAN | 2 | 1 | no | yes | 0.9 | 0.999 | 1 |
| 4-GAN rel. bg. | NS-GAN | none | WGAN | 1 | 2 | no | no | 0.9 | 0.999 | 1 |
| 5-GAN rel. bg. | NS-GAN | none | WGAN | 2 | 2 | no | no | 0.9 | 0.999 | 1 |

Table 1: The best hyper-parameter configuration for each model as were obtained after conducting a grid search. Configurations were chosen based on the smallest average FID (across 5 seeds).

# B    Experiment Details

## B.1    Model specifications

The generator and discriminator neural network architectures in all our experiments are based on DCGAN (Radford et al., 2015).

**Object Generators**    *k-GAN ind.* introduces $K = k$ copies of an object generator (i.e. tied weights, DCGAN architecture) that each generate and image from an independent sample of a 64-dimensional *UNIFORM(-1, 1)* prior $P(Z)$.

**Relational Structure**    When a relational stage is incorporated (*k-GAN rel.*) each of the $z_i \sim P(Z)$ are first updated, before being passed to the generators. These updates are computed using one or more *attention blocks* (Zambaldi et al., 2018), which integrate Multi-Head Dot-Product Attention (MHDPA; Vaswani et al. (2017)) with a post-processing step. A single head of an attention block updates $z_i$ according to (3).

In our experiments we use a single-layer neural network (fully-connected, 32 ReLU) followed by LayerNorm (Ba et al., 2016) for each of MLP$^{query}$, MLP$^{key}$, MLP$^{value}$. We implement MLP$^{up}$ with a two-layer neural network (each fully-connected, 64 ReLU), and apply LayerNorm after summing with $z_i$. Different heads in the same block use different parameters for MLP$^{query}$, MLP$^{key}$, MLP$^{value}$, MLP$^{up}$. If multiple heads are present, then their outputs are concatenated and transformed by a single-layer neural network (fully-connected, 64 ReLU) followed by LayerNorm to obtain the new $\hat{z}_i$. If the relational stage incorporates multiple attention blocks that iteratively update $z_i$, then we consider two variations: using unique weights for each MLP in each block, or sharing their weights across blocks.

**Background Generation**    When a background generator is incorporated (eg. *k-GAN rel. bg*) it uses the same DCGAN architecture as the object generators, yet maintains its own set of weights. It receives as input its own latent sample $z_b \sim P(Z_b)$, again using a *UNIFORM(-1, 1)* prior, although one may in theory choose a different distribution. We explore both variations in which $z_b$ participates in the relational stage, and in which it does not.

**Composing**    In order to obtain the final generated image, we need to combine the images generated by each generator. In the case of *Independent MM* and *Triplet MM* we simply sum the outputs of the different generators and clip their values to $(0, 1)$. On *RGB Occluded MM* we combine the different outputs using alpha compositing, with masks obtained by thresholding the output of each generator at $0.1$. On *CIFAR10 + MM* and *CLEVR* we require each of the object generators to generate an additional alpha channel by adding an additional feature map in the last layer of the generator. These are then combined with the generated background (opaque) using alpha compositing, i.e. through repeated application of (4).

## B.2    Hyperparameter Configurations

Each model is optimized with ADAM (Kingma & Ba, 2015) using a learning rate of $0.0001$, and batch size 64 for $1\,000\,000$ steps. Each generator step is followed by 5 discriminator steps, as is considered best practice in training GANs. Checkpoints are saved at every $20\,000^{th}$ step and we consider only the checkpoint with the lowest FID for each hyper-parameter configuration. FID is computed using $10\,000$ samples from a hold-out set.

**Baseline**    We conduct an extensive grid search over 48 different GAN configurations to obtain a strong GAN baseline on each dataset. It is made up of hyper-parameter ranges that were found to be successful in training GANs on standard datasets (Kurach et al., 2018).

We consider [SN-GAN / WGAN], using [NO / WGAN] gradient penalty with $\lambda$ [1 / 10]. In addition we consider these configurations [WITH / WITHOUT] spectral normalization. We consider [(0.5, 0.9) / (0.5, 0.999) / (0.9, 0.999)] as $(\beta_1, \beta_2)$ in ADAM. We explore 5 different seeds for each configuration.

**k-GAN**   We conduct a similar grid search for the GANs that incorporate our proposed structure. However, in order to maintain a similar computational budget to our baseline we consider a *subset* of the previous ranges to compensate for the additional hyper-parameters of the different structured components that we would like to search over.

In particular, we consider SN-GAN with WGAN gradient penalty, with a default $\lambda$ of 1, [WITH / WITHOUT] spectral normalization. We use (0.9, 0.999) as fixed values for ($\beta_1$, $\beta_2$) in ADAM. Additionally we consider K = [3 / 4 / 5] copies of the generator, and the following configurations for the relational structure:

- Independent
- Relational (1 block, no weight-sharing, 1 head)
- Relational (1 block, no weight-sharing, 2 heads)
- Relational (2 blocks, no weight-sharing, 1 head)
- Relational (2 blocks, weight-sharing, 1 head)
- Relational (2 blocks, no weight-sharing, 2 heads)
- Relational (2 blocks, weight-sharing, 2 heads)

This results in 42 hyper-parameter configurations, for which we each consider 5 seeds. We do not explore the use of a background generator on the non-background datasets. Correspondingly, we *only* explore variations that incorporate the background generator on the background datasets. In the latter case we search over an additional hyper-parameter that determines whether the latent representation of the background generator should participate in the relational stage or not.

### B.3   HUMAN STUDY

We asked human raters to compare the images generated by *k-GAN* ($k = 3, 4, 5$) to our baseline on *RGB Occluded MM*, *CIFAR10 + MM* and *CLEVR*, using the configuration with a background generator for the last two datasets. For each model we select the 10 best hyper-parameter configurations, from which we each generate 100 images. We conduct two different studies 1) in which we compare images from *k-GAN* against *GAN* and 2) in which we asked raters to answer questions about the content (properties) of the images.

**Comparison**   We asked reviewers to compare the quality of the generated images. We asked up to three raters for each image and report the majority vote or "none" if no decision can be reached.

**Properties**   For each dataset we asked (up to three raters for each image) the following questions.

On *RGB Occluded MM* we asked:

1. How many [red, blue, green] shapes are in the image? Answers: [0, 1, 2, 3, 4, 5]
2. How many are recognizable as digits? Answers: [0, 1, 2, 3, 4, 5]
3. Are there exactly 3 digits in the picture, one of them green, one blue and one red? Answers: Yes / No

On *CIFAR10 + MM* we asked these same questions, and additionally asked:

4. Does the background constitute a realistic scene? Answers: Yes / No

On *CLEVR* we asked the following set of questions:

1. How many shapes are in the image? Answers: [0, 1, 2, 3, 4, 5, 6, 7, 8, 9, 10]
2. How many are recognizable as geometric objects? Answers: [0, 1, 2, 3, 4, 5, 6, 7, 8, 9, 10]
3. Are there any objects with mixed colors (eg. part green part red)? Answers: Yes / No
4. Are there any objects with distorted geometric shapes?: Answers: Yes / No
5. Are there any objects that appear to be floating? Answers: Yes / No
6. Does the scene appear to be crowded? Answers: Yes / No

## C    OVERVIEW OF REAL AND GENERATED SAMPLES

Generated samples are reported for the best (lowest FID) structured model, as well as the best baseline model for each dataset.

### C.1    INDEPENDENT MULTI MNIST

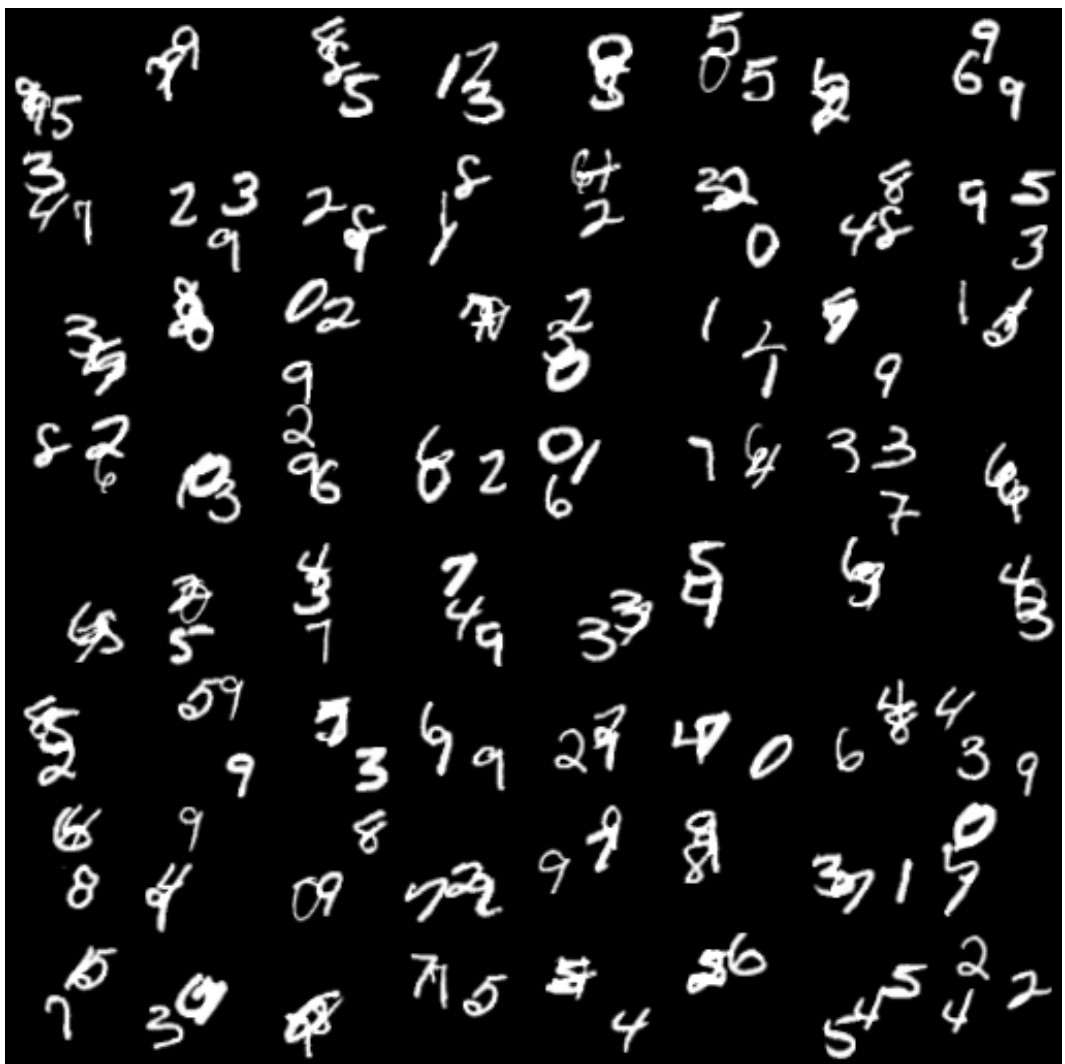

Figure 10: Real

### C.1.1 STRUCTURED GAN

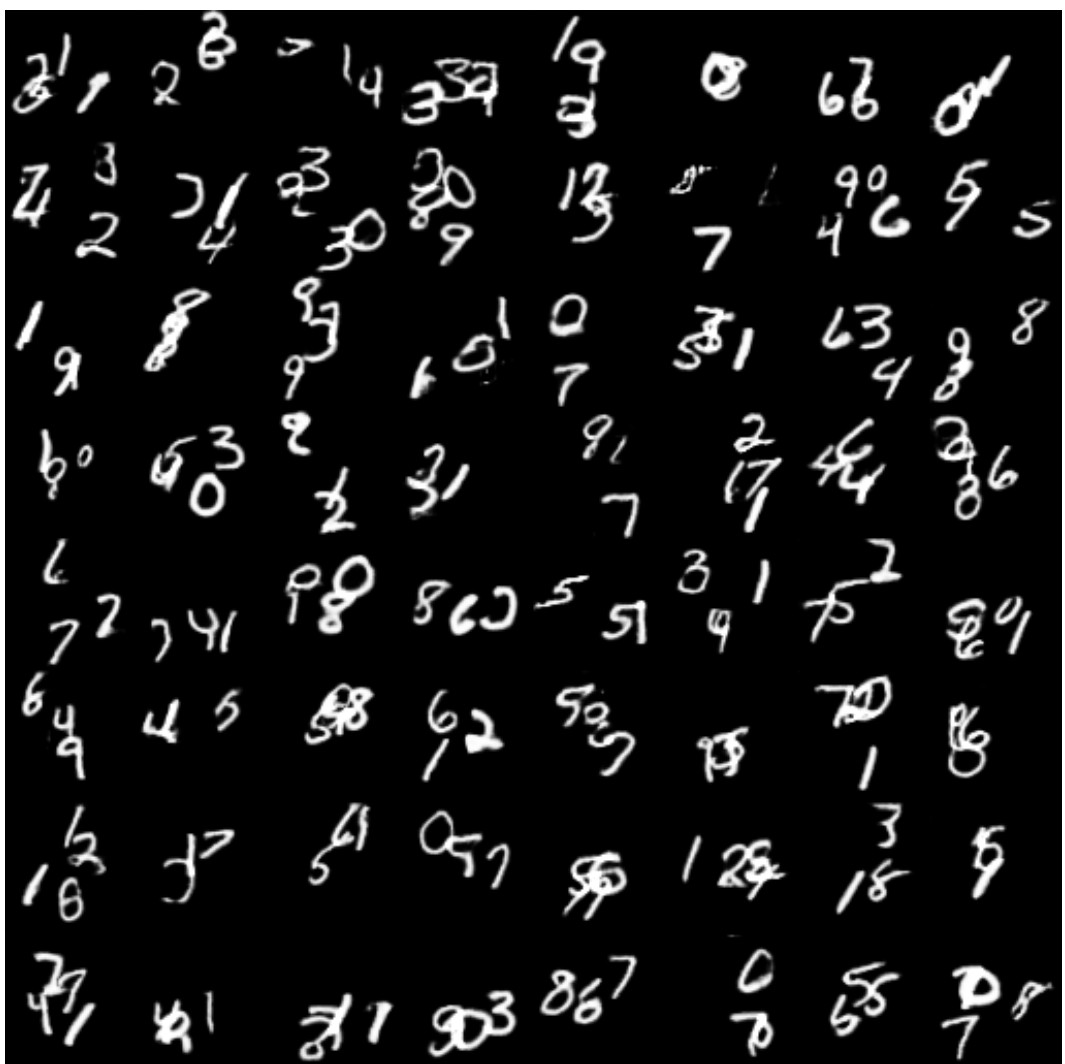

Figure 11: 4-GAN ind. with spectral norm and WGAN penalty

### C.1.2  GAN

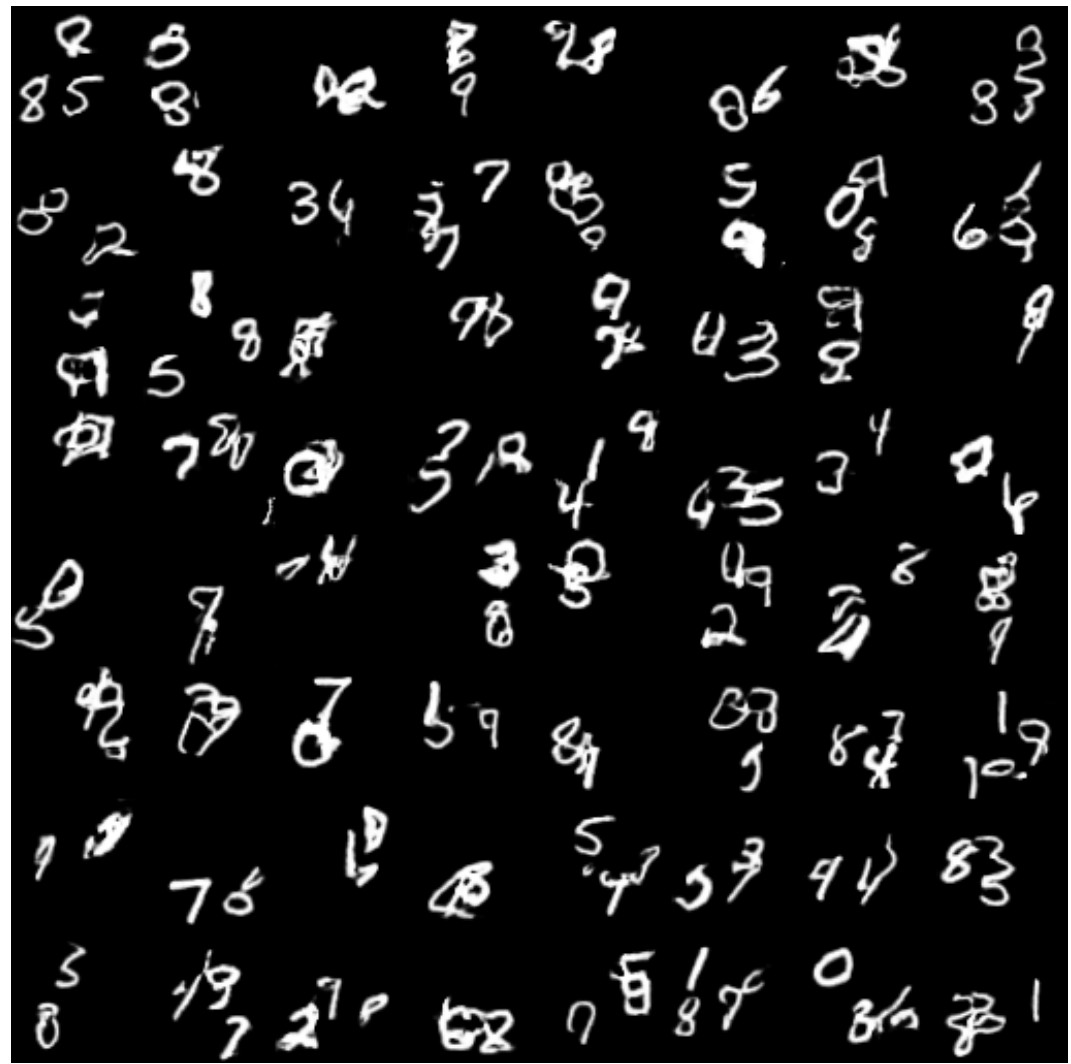

Figure 12: NS-GAN with spectral norm

## C.2 TRIPLET MULTI MNIST

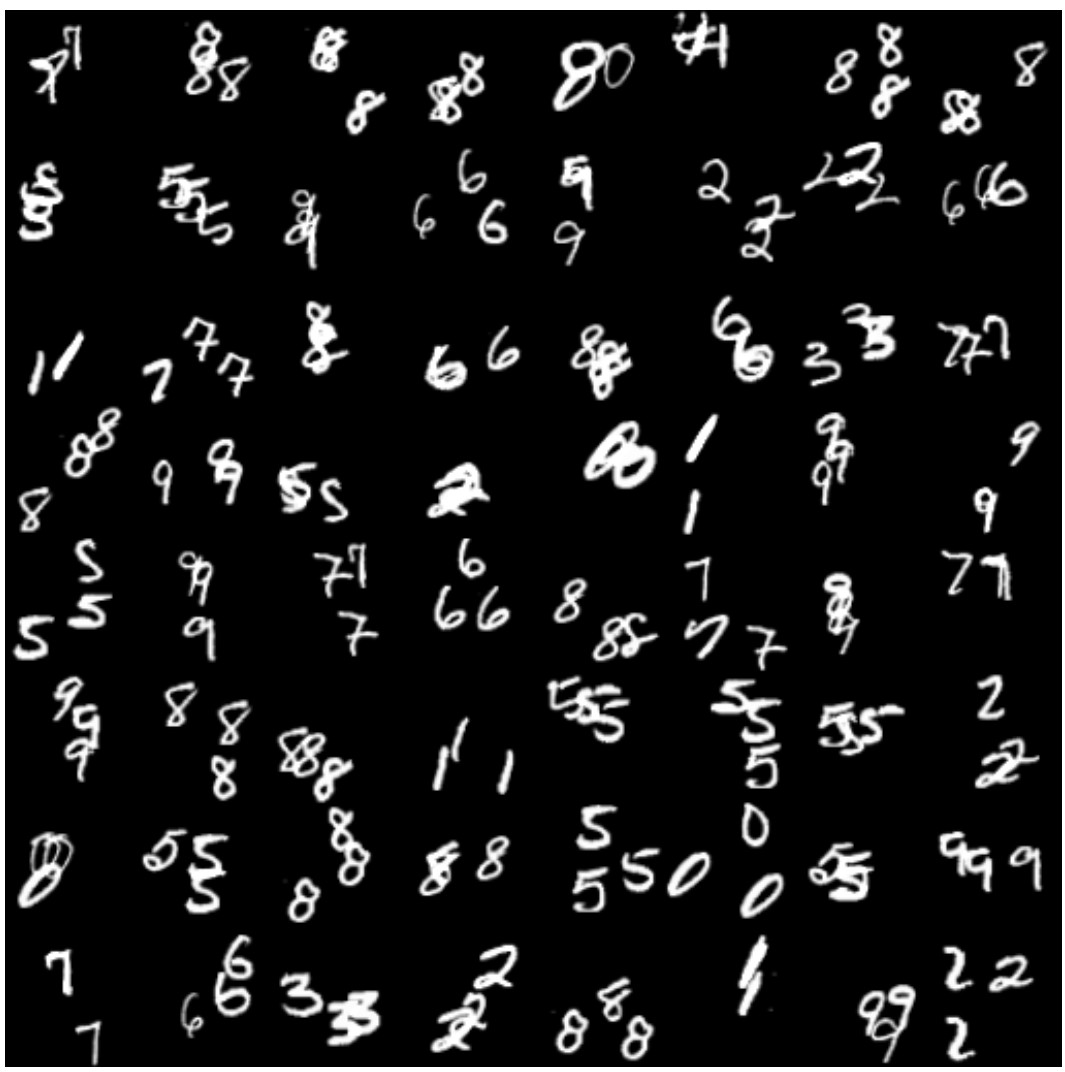

Figure 13: Real

### C.2.1 STRUCTURED GAN

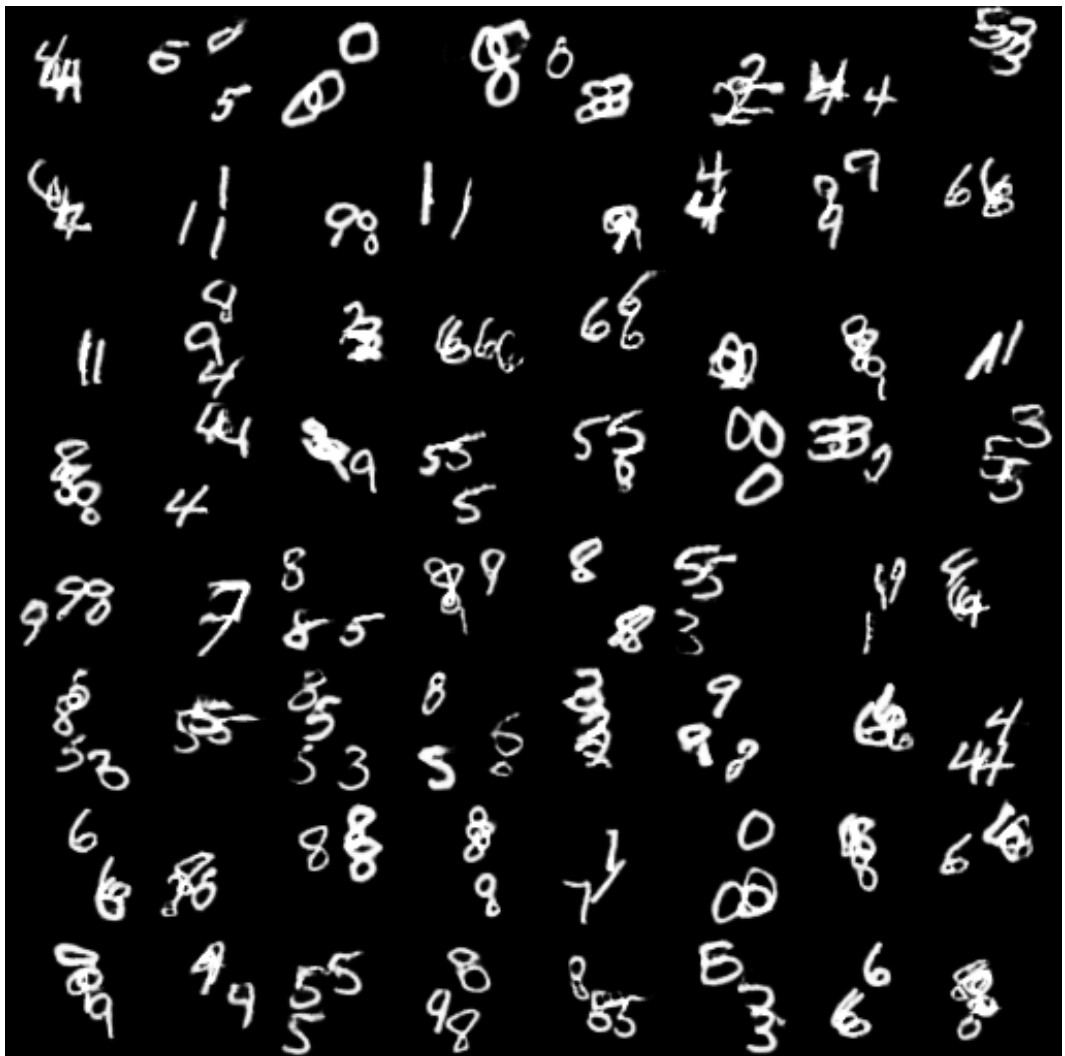

Figure 14: 4-GAN rel. (1 block, 2 heads, no weight sharing) with spectral norm and WGAN penalty

### C.2.2 GAN

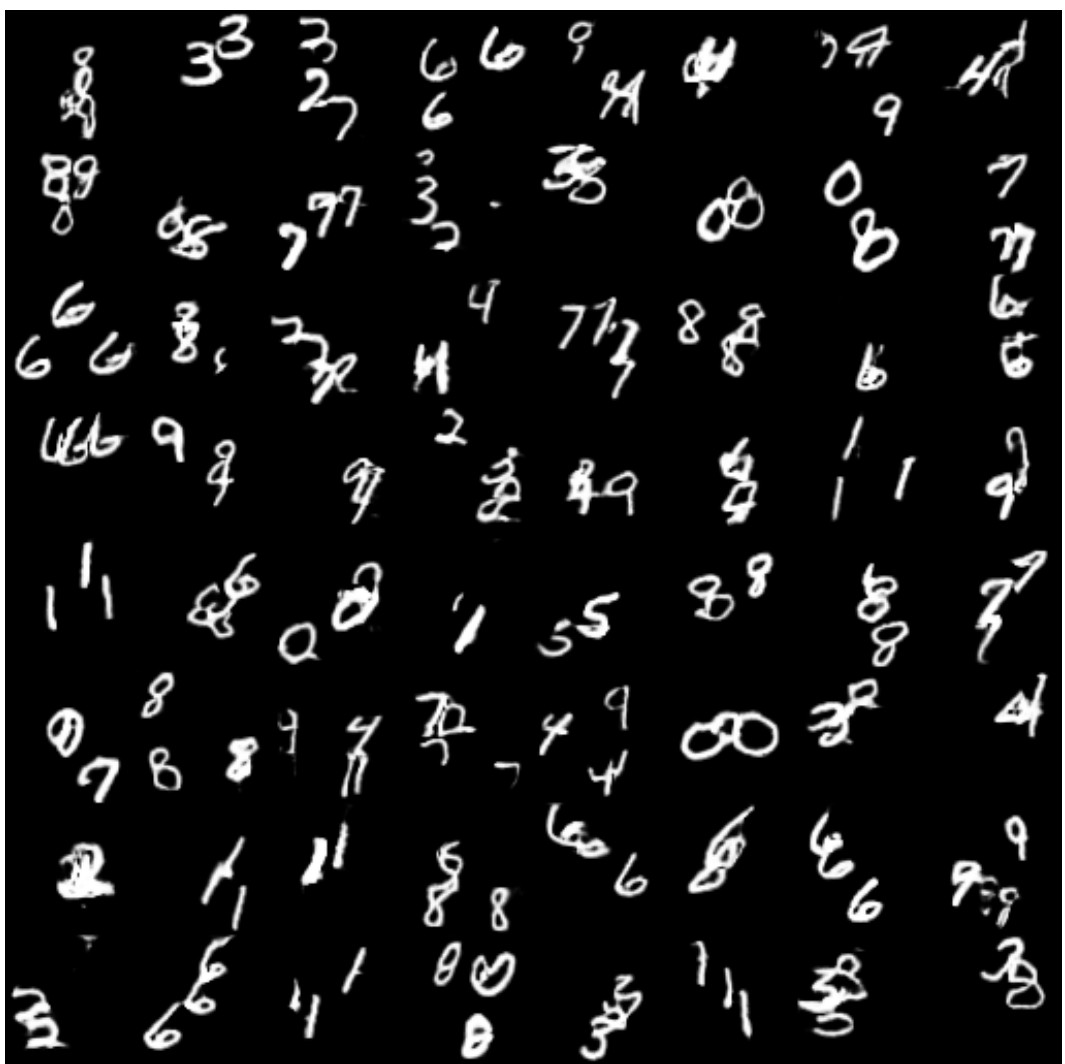

Figure 15: NS-GAN with spectral norm

## C.3 RGB-OCCLUDED MULTI MNIST

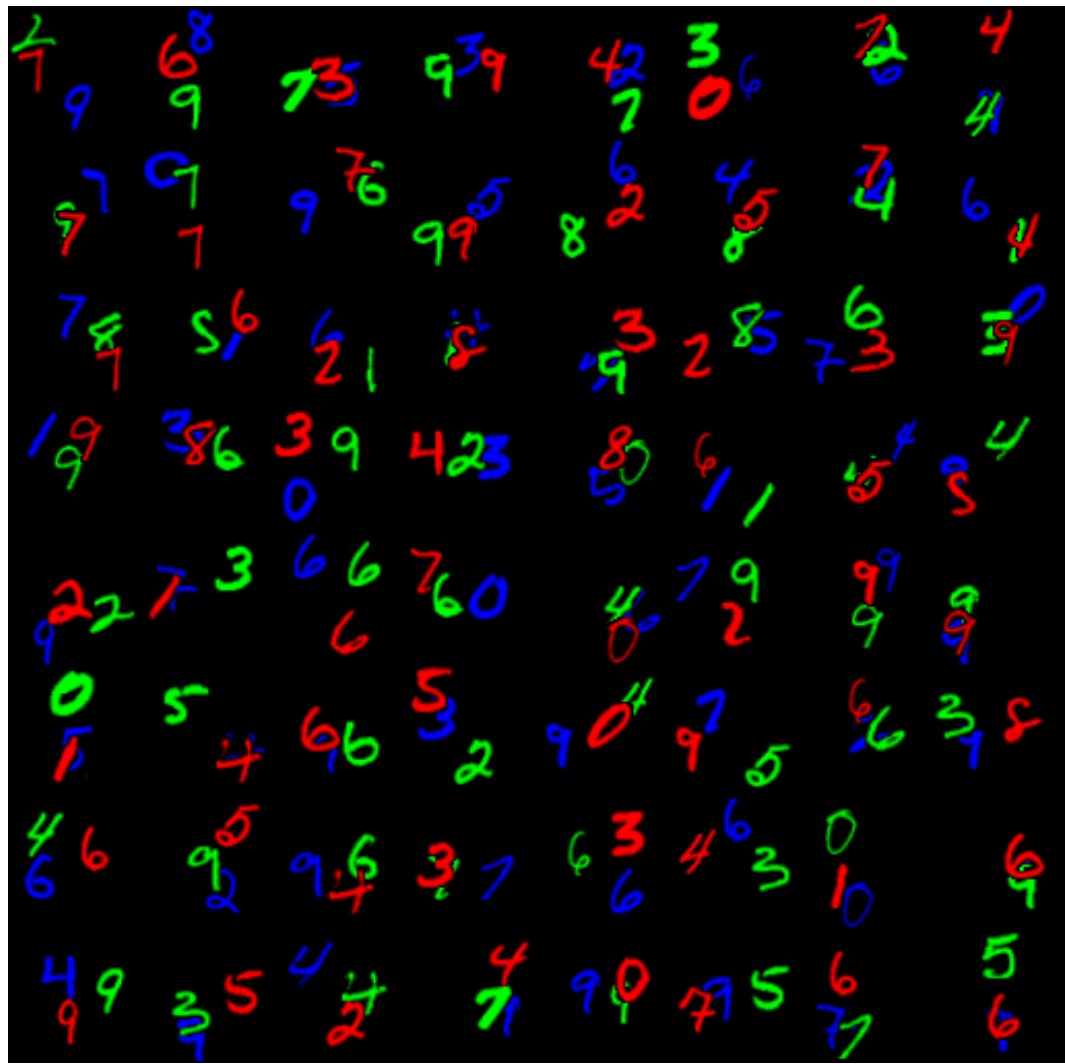

Figure 16: Real

## C.3.1 STRUCTURED GAN

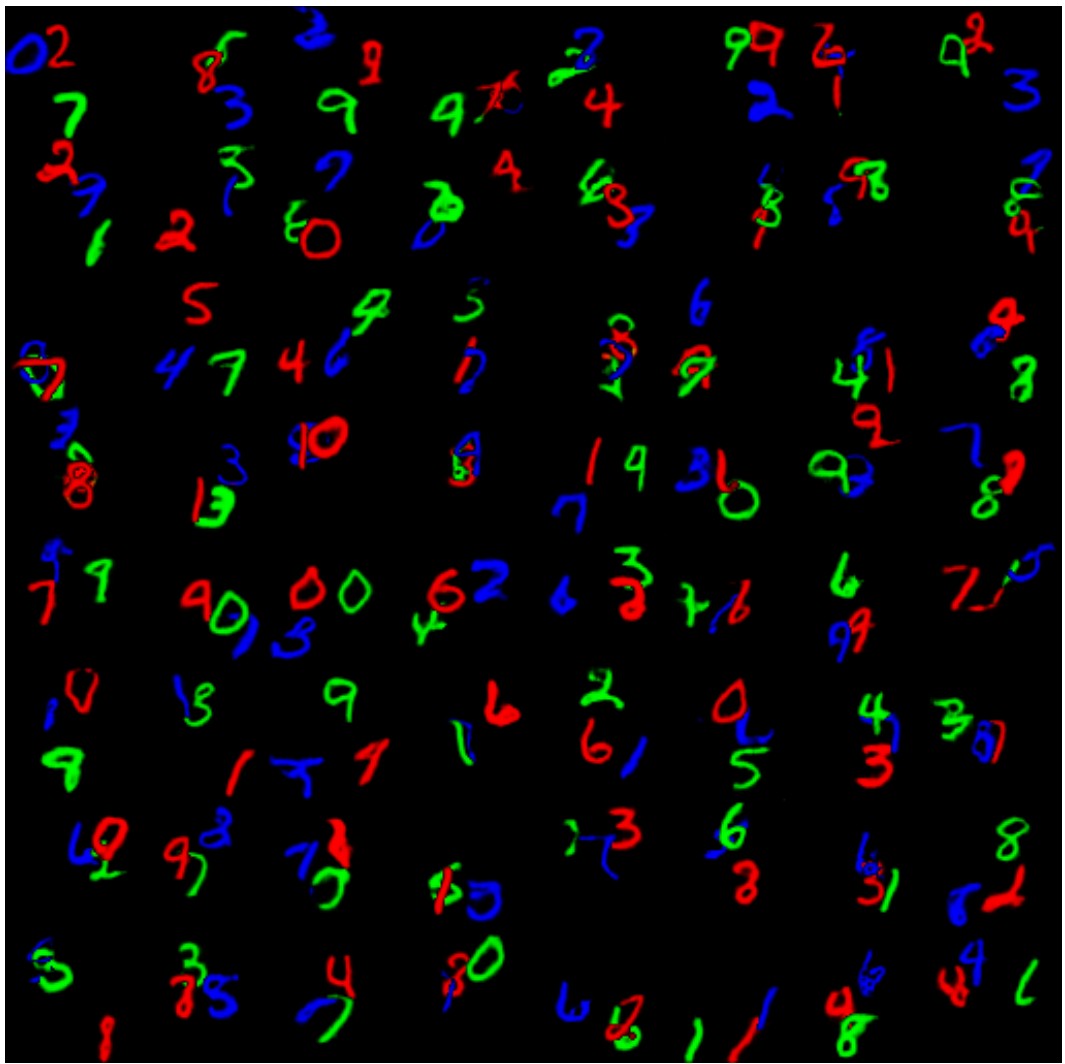

Figure 17: 3-GAN rel. (2 blocks, 2 heads, no weight sharing) with spectral norm and WGAN penalty

C.3.2   GAN

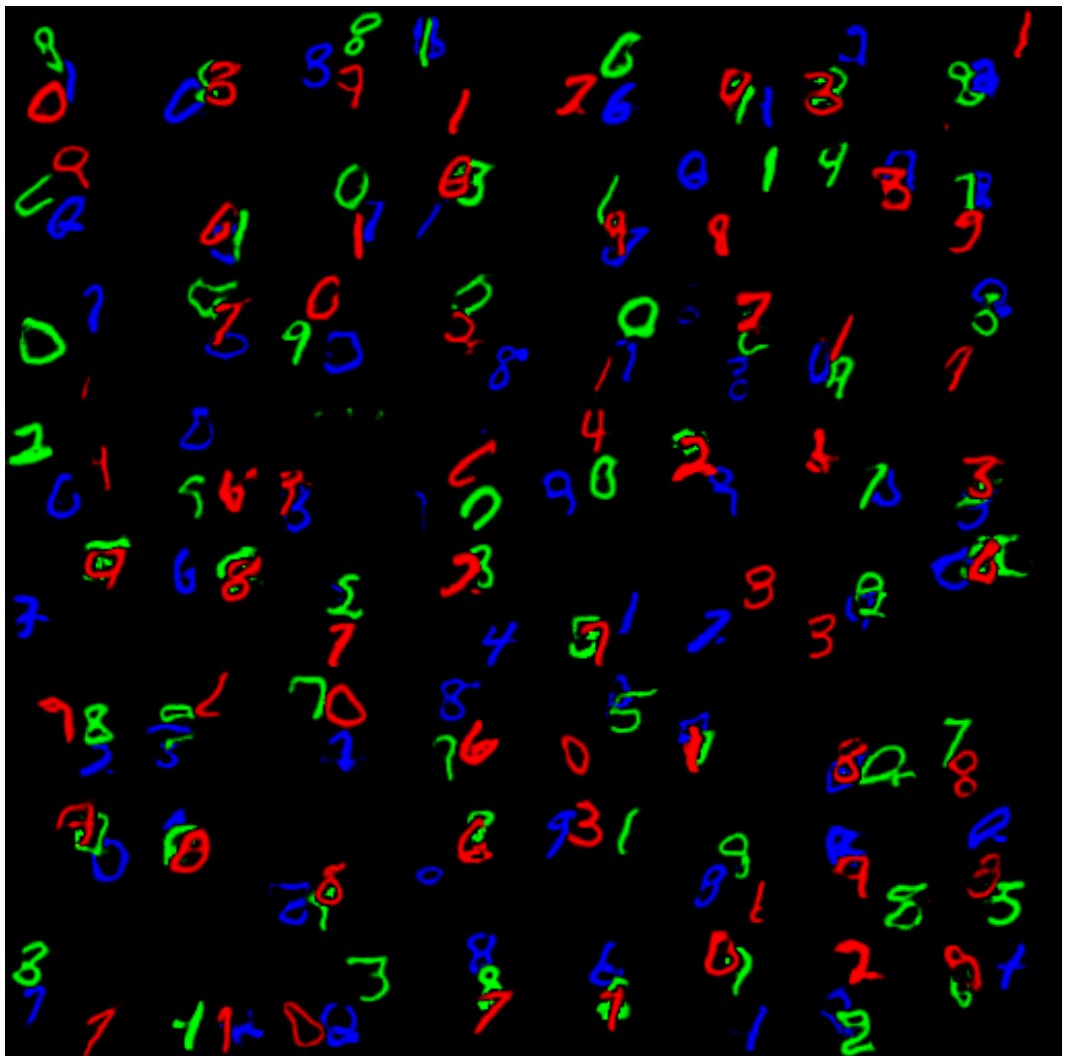

Figure 18: NS-GAN with spectral norm

## C.4    RGB-Occluded Multi MNIST + CIFAR10

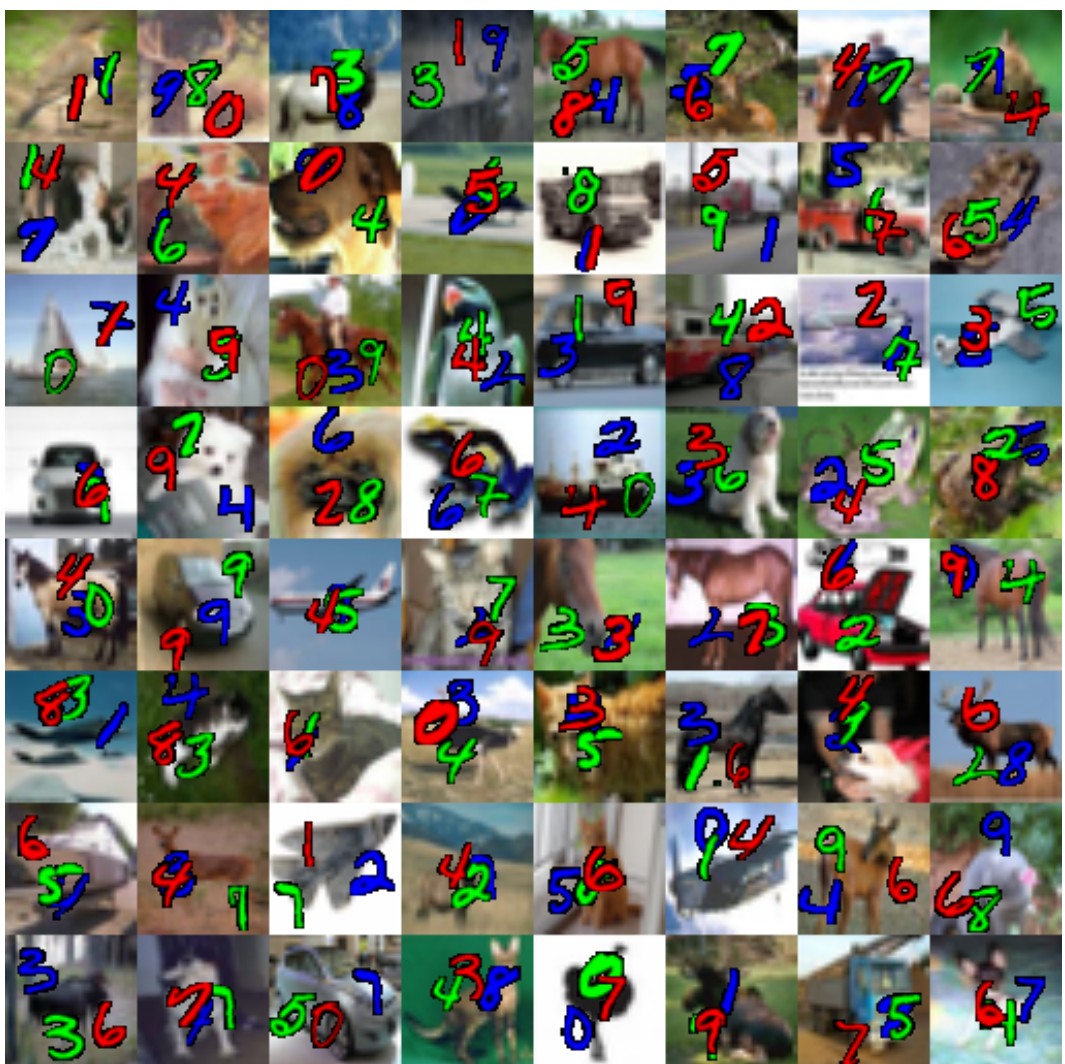

Figure 19: Real

### C.4.1 STRUCTURED GAN

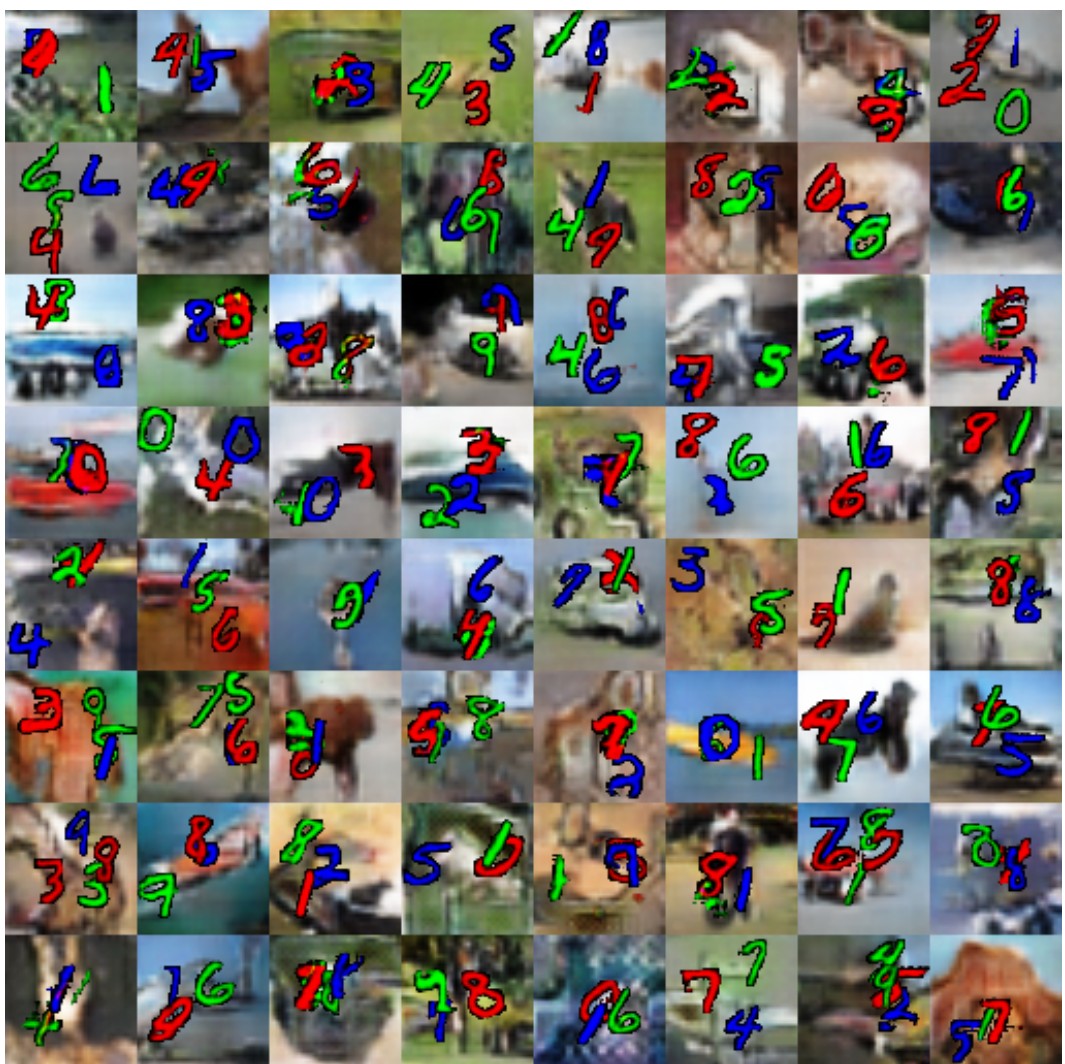

Figure 20: 5-GAN rel. (1 block, 2 heads, no weight sharing) bg. (no interaction) with WGAN penalty

### C.4.2  GAN

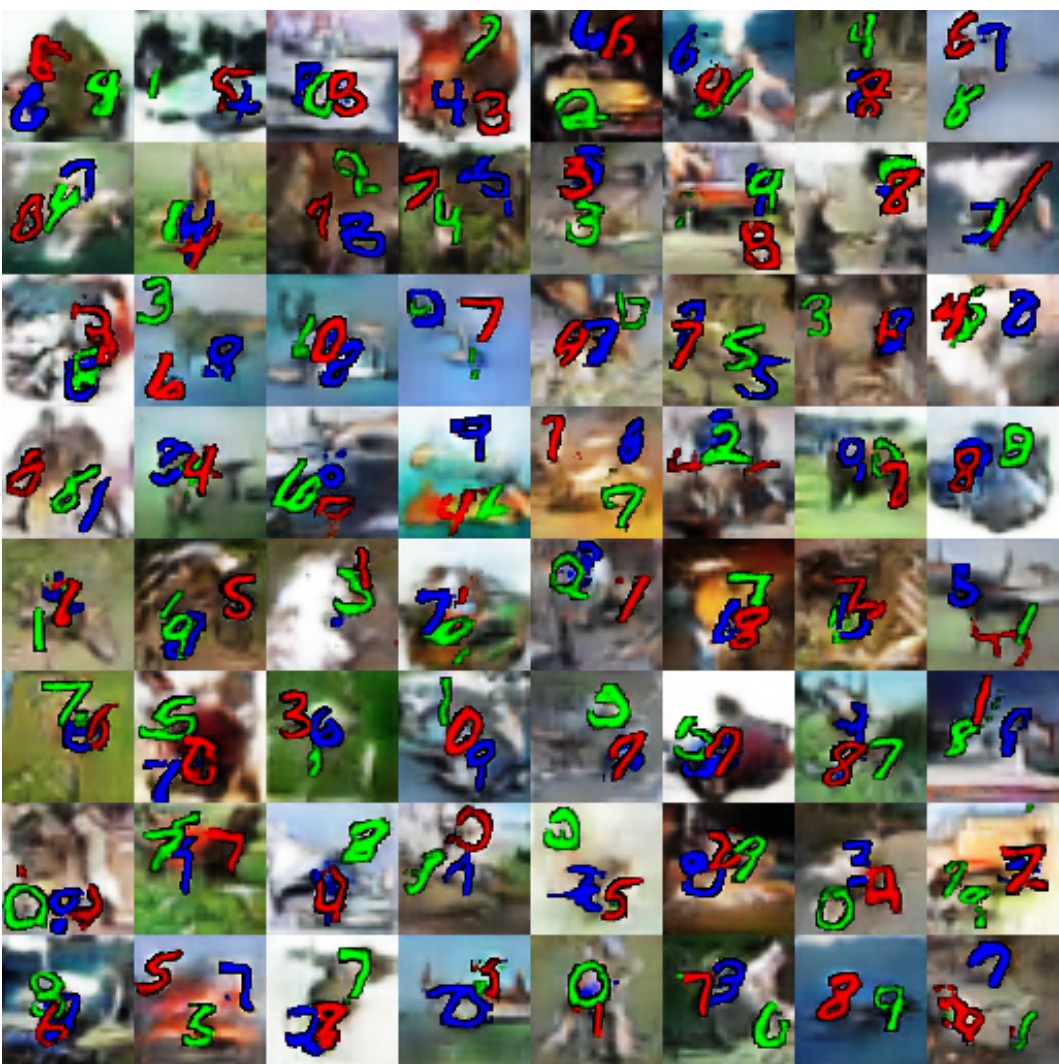

Figure 21: WGAN with WGAN penalty

## C.5 CLEVR

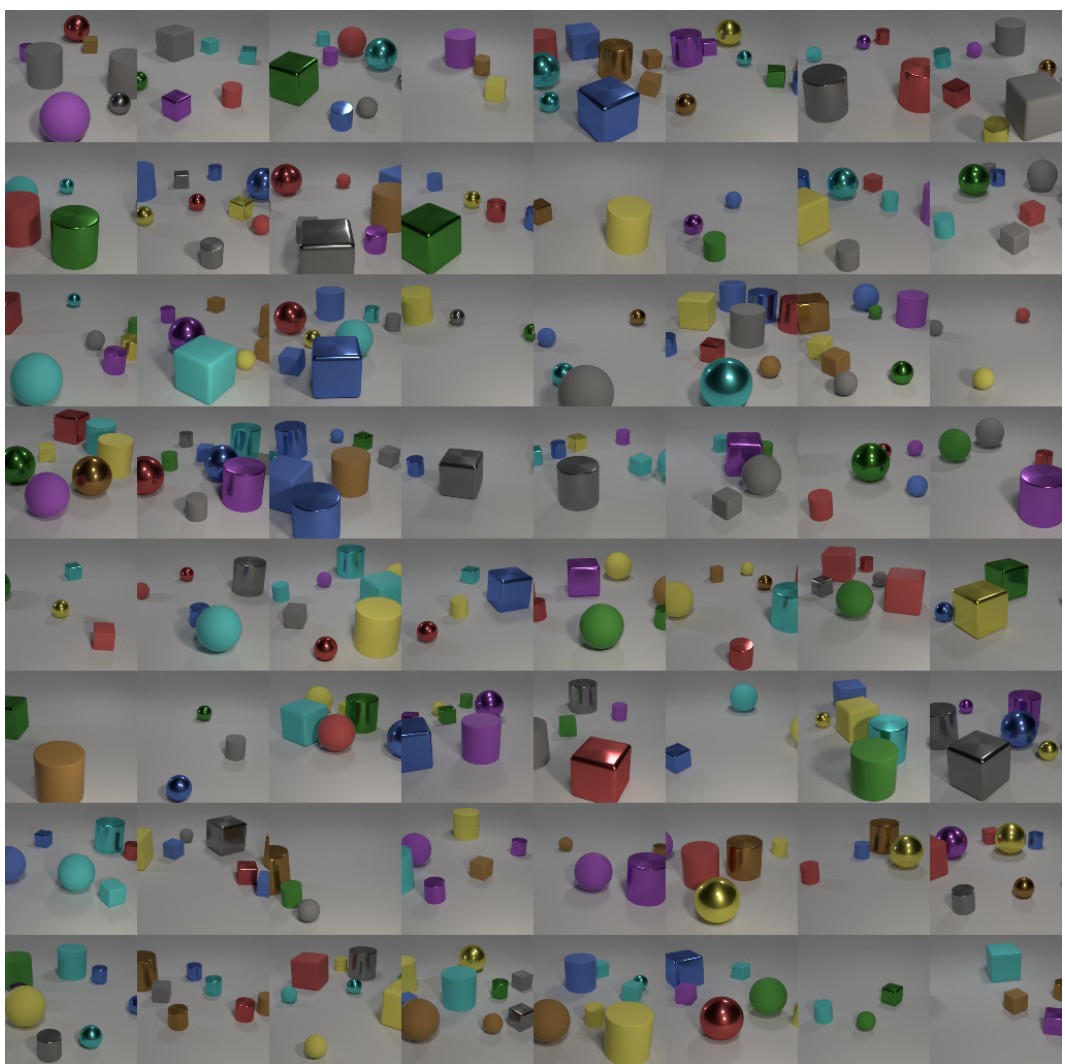

Figure 22: Real

### C.5.1 Structured GAN

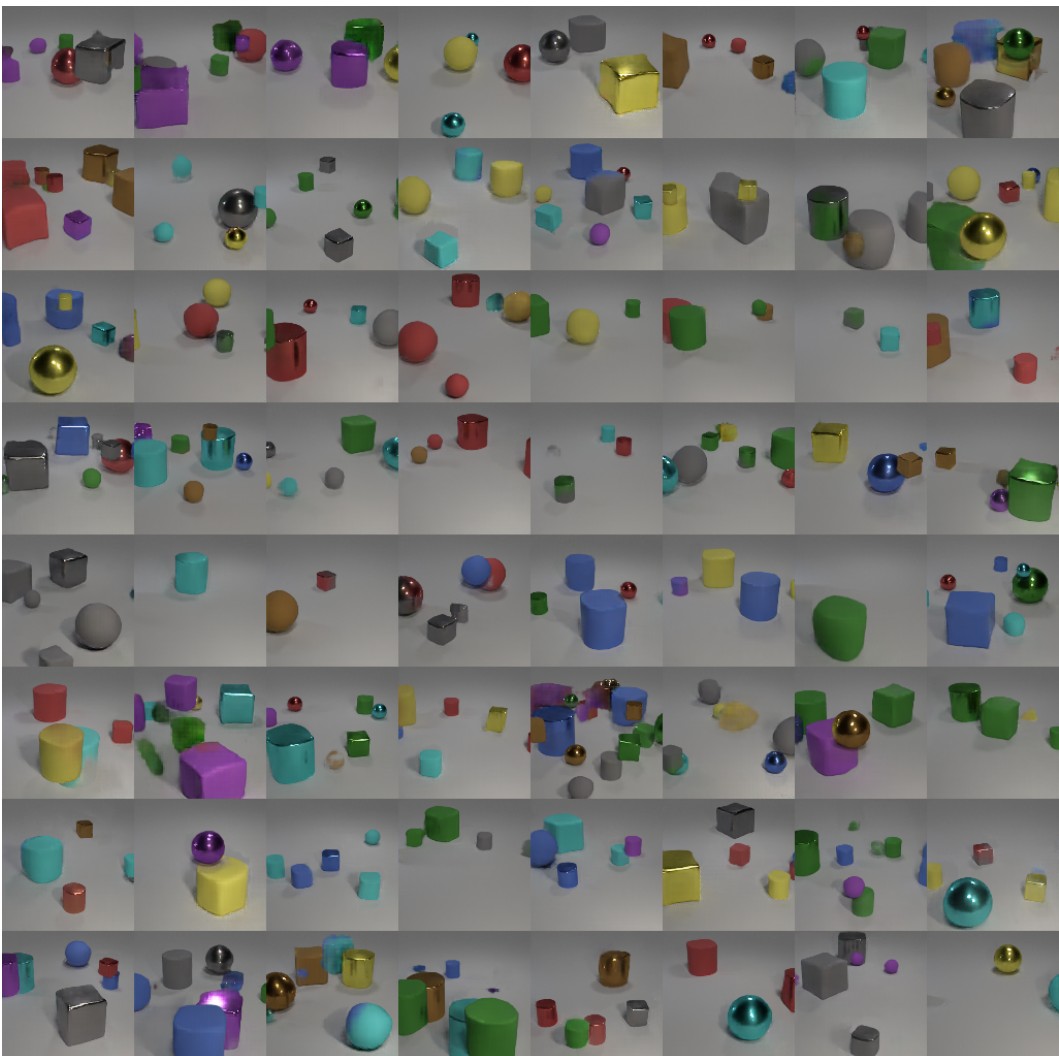

Figure 23: 3-GAN rel. (2 heads, 2 blocks, no weight sharing) bg. (with interaction) with WGAN penalty.

### C.5.2 GAN

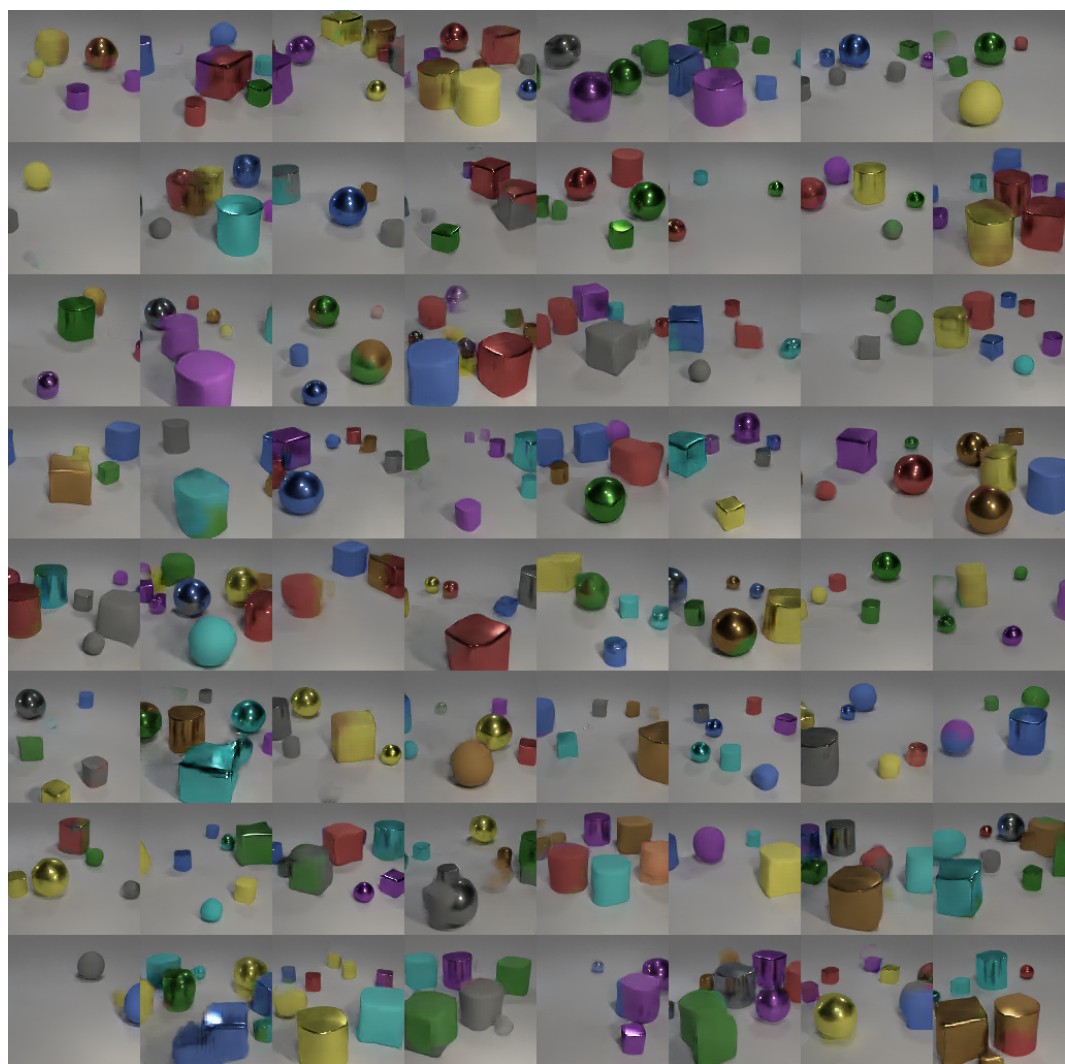

Figure 24: WGAN with WGAN penalty

