# OpenReview forum: "A Case for Object Compositionality in Deep Generative Models of Images"
_ICLR.cc/2019/Conference_

### Official Review · AnonReviewer3 · 2018-11-02
**Interesting idea but with insufficient experimental validation**

**Rating:** 6
**Confidence:** 4

**Review:**

The paper proposes a compositional generative model for GANs. Basically, assuming existence of K objects in the image, the paper creates a latent representation for each object as well as a latent representation for the background. To model the relation between objects, the paper utilizes the multi-head dot-product attention (MHDPA) due to Vaswani et a. 2017. Applying MHDPA to the K latent representations results in K new latent representations. The K new representations are then fed into a generator to synthesize an image containing one object. The K images together with the synthesized background image are then combined together to form the final image. The paper compares to the proposed approach to the standard GAN approach. The reported superior performance suggest the inductive bias of compositionality of scene leads to improved performance.

The method presented in the paper is a sensible approach and is overall interesting. The experiment results clearly shows the advantage of the proposed method. However, the paper does have several weak points. Firs of all, it misses an investigation of alternative network design for achieving the same compositionality. For example, what would be the performance difference if one replace the MHDPA with LSTM. Another weak point is that it is unclear if the proposed method can be generalize to handle more complicated scene such as COCO images as the experiments are all conducted using very toy-like image datasets.

---

> ### Author Response · Authors · 2018-11-16
> **Reply to Reviewer 3**
>
> Thank you for your consideration and feedback.
>
> The primary motivation of this work is to argue for object compositionality in deep generative models (and in particularly GANs), which originates from two key observations. First, real-world images are to a large degree compositional, and a generative model that is suitable equipped with a corresponding inductive bias should be better at capturing this distribution. Second, in disentangling information content corresponding to different objects at a representational level they may be recovered a posteriori unlike in unstructured models.
>
>  > “... it is unclear if the proposed method can be generalize to handle more complicated scene such as COCO images as the experiments are all conducted using very toy-like image datasets.”
>
> Our works builds on prior work in purely unsupervised multi-object image generation and representation learning. Whereas prior work has focused primarily on the representation learning part (eg. [3, 4, 5]), here our focus is on scaling these ideas to more complex datasets. In particular, among the multi-object datasets considered in relevant prior work are Multi-MNIST [3, 4, 7], Shapes [4, 5], and Textured MNIST [5]. In this work we consider several more complex datasets, including two relational version of Multi-MNIST (triplet, rgb), a variation on CIFAR10 that has RGB MNIST digits in the foreground, and high-resolution CLEVR images that contain many rendered geometric objects and require lighting and shadows to be modeled.
>
> Ideally we would be able to apply our approach to common segmentation datasets (eg. Pascal VOC, COCO) although in practice we find that these are still far out of reach for purely unsupervised approaches. Such datasets have been designed with access to ground-truth labels in mind and the large imbalance between the visual complexity of objects (i.e. intra-class variation) and the number of samples renders them unsuitable for our purpose. We consider CLEVR to be among the more complex multi-object datasets that are balanced in this way, and hence the feasibility of our approach on this dataset is an important step forward compared to prior work.
>
> > “... it misses an investigation of alternative network design for achieving the same compositionality. For example, what would be the performance difference if one replace the MHDPA with LSTM. “
>
> Our proposed framework incorporates MHDPA to model relations between objects. MHDPA is an instance of a graph network [1], which renders it suitable for this task. It would be valid to compare MHDPA to other instances of graph networks, eg. the interaction function from [6] or the relational mechanism from [2]. In prior experiments we have explored several ablations and extensions of the current relational mechanism that approach these configurations. We were unable to obtain significantly better FID scores for any of these variations, and so we settled with the mechanisms proposed in [8] to model relations between objects. As our goal was to demonstrate the feasibility / benefits of incorporating such a mechanisms we did not consider it worth it to dedicate human evaluation to this. However, we do agree that the paper could mention this and we will update it accordingly to make this more clear.
>
> [1] Battaglia, Peter W., et al. "Relational inductive biases, deep learning, and graph networks." arXiv preprint arXiv:1806.01261 (2018).
> [2] Chang, Michael B., et al. "A compositional object-based approach to learning physical dynamics." International Conference on Learning Representations. 2016.
> [3] Eslami, SM Ali, et al. "Attend, infer, repeat: Fast scene understanding with generative models." Advances in Neural Information Processing Systems. 2016.
> [4] Greff, Klaus, et al. "Neural expectation maximization." Advances in Neural Information Processing Systems. 2017.
> [5] Greff, Klaus, et al. "Tagger: Deep unsupervised perceptual grouping." Advances in Neural Information Processing Systems. 2016.
> [6] van Steenkiste, Sjoerd, et al. "Relational neural expectation maximization: Unsupervised discovery of objects and their interactions." International Conference on Learning Representations. 2018.
> [7] Yang, Jianwei, et al. "LR-GAN: Layered recursive generative adversarial networks for image generation." International Conference on Learning Representations. 2017.
> [8] Zambaldi, Vinicius, et al. "Relational Deep Reinforcement Learning." arXiv preprint arXiv:1806.01830 (2018).

---

### Official Review · AnonReviewer1 · 2018-11-03
**Interesting idea but not novel and ultimately unconvincing**

**Rating:** 4
**Confidence:** 5

**Review:**

This paper explores compositional image generation. Specifically, from a set of latent noises, the relationship between the objects is modelled using an attention mechanism to generate a new set of latent representations encoding the relationship. A generator then creates objects separately from each of these (including alpha channels). A separate generator creates the background. The objects and background are finally combined in a final image using alpha composition. An independent setting is also explored, where the objects are directly sampled from a set of random latent noises.

My main concern is that the ideas, while interesting, are not novel, the method not clearly motivated, and the paper fails to convince.

It is interesting to see that the model was able to somewhat disentangle the objects from the background. However, overall, the experimental setting is not fully convincing. The generators seem to generate more than one object, or backgrounds that do contain objects. The datasets, in particular, seem overly simplistic, with background easily distinguishable from the objects. A positive point is that all experimented are ran with 5 different seeds. The expensive human evaluation used does not provide full understanding and do not seem to establish the superiority of the proposed method.

The very related work by Azadi et al on compositional GAN, while mentioned, is not sufficiently critiqued or adequately compared to within the context of this work.

The choice of an attention mechanism to model relationship seems arbitrary and perhaps overly complicated for simply creating a set of latent noises. What happens if a simple MLP is used? Is there any prior imposed on the scene created? Or on the way the objects should interact?
On the implementation side, what MLP is used, how are its parameters validated?

What is the observed distribution of the final latent vectors? How does this affect the generation process? Does the generator use all the latent variables or only those with highest magnitude?
The attention mechanism has a gate, effectively adding in the original noise to the output — is this a weighted sum? If so, how are the coefficient determined, if not, have the authors tried?

The paper goes over the recommended length (still within the limit) but still fails to include some important details —mainly about the implementation— while some of the content could be shortened or moved to the appendix. Vague, unsubstantiated claims, such as that structure of deep generative models of images is determined by the inductive bias of the neural network are not really explained and do not bring much to the paper.

---

> ### Author Response · Authors · 2018-11-16
> **Reply to Reviewer 1 [1/ 3]**
>
> Thank you for your feedback and consideration.
>
> In the following we first provide an overview of the main answers regarding your main concern that “... the ideas, while interesting, are not novel, the method not clearly motivated, and the paper fails to convince” before proceeding to a detailed discussion:
>
> * Compositionality is critical in reducing complex visual scenes to a set of primitives (objects) that can be re-combined freely to generate new scenes (combinatorial productivity). There is substantial empirical evidence that neural networks can benefit from this in image processing tasks.
> * The novelty of our approach is in combining insights from unsupervised multi-object image processing (representation learning) with GANs that have proven useful in generating complex images.
> * The datasets that we consider are non-trivial and substantially more complex compared to relevant related work that only considers variations of Multi-MNIST. Our results on CLEVR already significantly advance upon the state of the art in terms of unsupervised multi-object image-processing.
> * The experimental evaluation is sound and the reported results are representative for the model performance: it consistently generates images as a composition of individual objects.
> * Compared to a strong baseline of GANs we find that the generated images are of higher quality, and more faithful to the reference distribution, as confirmed by a large scale human study.
>
> Detailed answers below:
>
> The primary motivation of this work is to argue for object compositionality in deep generative models (and in particularly GANs), which originates from two key observations. First, real-world images are to a large degree compositional, and a generative model that is suitable equipped with a corresponding inductive bias should be better at capturing this distribution. Second, in disentangling information content corresponding to different objects at a representational level they may be recovered a posteriori unlike in unstructured models.
>
> In our experiments we find that the proposed model is successful in doing both: it generates images of higher-quality that are more faithful to the reference distribution (as per human evaluation), and it consistently disentangles information content belonging to different objects (visual inspection).
>
> We would like to emphasize this last part, and point out that the reported images in Figures 3 & 4 are representative of how the best performing models generate the scene. In other words, on all datasets we consistently find that the network generates the images as a composition of individual objects. Indeed on CLEVR we found cases in which a component generates more than one object, which is understood by the fact that the number of components is smaller than the number of objects in the scene (see also Figure 8 in Appendix A). We also find infrequent cases (primarily on CLEVR, although sometimes on CIFAR10) in which the background generator generates an additional object. This is understandable as there are no restrictions in our approach that prevent it from doing so. However, given that in almost all other cases the network generate images as compositions of objects and background it seems reasonable to conclude that these are due to optimization issues (as are common in GANs). After all, the compositional solution is clearly the superior choice, as is evident from our human evaluation compared to regular GAN.
>
> The proposed framework combines insights from related work in multi-object image generation and relational reasoning. There is substantial evidence that object compositionality is beneficial in a variety of image-related tasks, although purely unsupervised approaches (in particular those targeted at discovering object representations) have only been evaluated on toy datasets. GANs have been shown to scale to complex images, and the proposed approach demonstrates that a combination of these ideas is fruitful. More specifically, our contributions are (1) an implementation of recent insights from unsupervised multi-object image processing in the GAN framework, (2) strong evidence that a deep generative model may learn about objects purely through the process of generation (i.e. without a “decoder”), (3) strong evidence that object compositionality benefits the quality and properties of generated images, and (4) strong evidence that these ideas can be scaled to more complex datasets in using GANs.

---

> > ### Author Response · Authors · 2018-11-16
> > **Reply to Reviewer 1 [2 / 3]**
> >
> > Regarding (4), we would like to point out that relevant prior work concerned with multi-object images focuses on Multi-MNIST [4, 5, 8], Shapes [5, 6], and Textured MNIST [6]. In this work we consider several more complex datasets, including two relational version of Multi-MNIST (triplet, rgb), a variation on CIFAR10 that has RGB MNIST digits in the foreground, and high-resolution CLEVR images that contain many rendered geometric objects and require lighting and shadows to be modeled.
> >
> > Ideally we would be able to apply our approach to common segmentation datasets (eg. Pascal VOC, COCO) although in practice we find that these are still far out of reach for purely unsupervised approaches. Such datasets have been designed with access to ground-truth labels in mind and the large imbalance between the visual complexity of objects (i.e. intra-class variation) and the number of samples renders them unsuitable for our purpose. We consider CLEVR to be among the more complex multi-object datasets that are balanced in this way, and hence the feasibility of our approach on this dataset is an important step forward compared to prior work.
> >
> > > “The very related work by Azadi et al on compositional GAN, while mentioned, is not sufficiently critiqued or adequately compared to within the context of this work.”
> >
> > Our approach is only marginally related to the Compositional GAN proposed by Azadi et al. [1]. Their approaches takes as input a pair of images (conditional generation) and corresponding segmentation masks that indicate which pixels of an input image belong to an object. Their framework then implements a means to compose the objects in the individual images to obtain a new image. On the other hand our generator only receives noise as input, and an important challenge is in learning how to disentangle information content belonging to different objects (identifying what objects are in the process), such that scenes may be generated in a compositional fashion. In that sense, our work and the work by Azadi et al. could be combined by using the Compositional GAN as a replacement for the “composition” operation in our approach, while ignoring the relational structure. One interesting observation is that the self-consistency loss from Azadi et al. could be used to learn a network that decomposes the composed image into images of individual objects, which is expected to benefit the discriminator (as per our discussion). We will update the discussion section to list this as a possibility for future work.
> >
> > > “The choice of an attention mechanism to model relationship seems arbitrary and perhaps overly complicated for simply creating a set of latent noises. What happens if a simple MLP is used?”
> >
> > The role of the attention mechanism in this work is to model object-object interactions, which motivates its choice. MHDPA is an instance of a graph network [2], and similar to other instances (eg. the interaction function in [7], or the relational mechanism in [3]) it excels at relational reasoning. In particular, by factorizing complex relations into pairwise interactions, and weight sharing, this mechanism is compositional and invariant in the number of objects. In prior experiments we have explored several abilations and extensions of the current relational mechanism that approach these configurations. We were unable to obtain significantly better FID scores for any of these variations, and so we settled with the mechanisms proposed in [9] to model relations between objects.

---

> > > ### Author Response · Authors · 2018-11-16
> > > **Reply to Reviewer 1 [3 / 3]**
> > >
> > > > “Is there any prior imposed on the scene created? Or on the way the objects should interact?”
> > >
> > > there is no prior imposed on the scene created (other than that is compositional) or in the way objects are supposed to interact.
> > >
> > > > “On the implementation side, what MLP is used, how are its parameters validated?”
> > >
> > > All implementation details are available in Appendix B.1 (layer sizes, normalization, activation functions etc.) The choice for this particular configuration of the relational mechanism were obtained after exploring several other variations that did not result in a significant improvement in FID scores. We agree that this is currently unclear in the paper and we will update the Appendix to reflect this. All other hyper-parameters listed in Appendix B.2 participate in a large-scale grid search in which we explore more than 250 different configurations, including 5 seeds.
> > >
> > > > “The attention mechanism has a gate, effectively adding in the original noise to the output — is this a weighted sum? If so, how are the coefficient determined, if not, have the authors tried?”
> > >
> > > As per equation (3) in the main text and appendix B.1, the update vector a_i is obtained as a weighted sum of the value vectors of each component. Attention weights are obtained by computing an inner product between the query vector q_i and the key vector k_i, followed by normalization and a softmax activation which ensures that the weights in the total sum add up to 1. The update vector a_i is passed through a post-processing network (2 layer MLP - see appendix for details) before being added to z_i (without a gate). We have not tried a configuration that gates the update a_i with z_i, in order to prevent the initial sample from being ignored.
> > >
> > > > “The paper goes over the recommended length (still within the limit) but still fails to include some important details —mainly about the implementation”
> > >
> > > It is our understanding that all experiment details (and other important details) are available in the paper. However, if you find that anything is unclear or missing, then we are happy to update the paper accordingly.
> > >
> > > [1] Azadi, Samaneh, et al. "Compositional GAN: Learning Conditional Image Composition." arXiv preprint arXiv:1807.07560 (2018).
> > > [2] Battaglia, Peter W., et al. "Relational inductive biases, deep learning, and graph networks." arXiv preprint arXiv:1806.01261 (2018).
> > > [3] Chang, Michael B., et al. "A compositional object-based approach to learning physical dynamics." International Conference on Learning Representations. 2016.
> > > [4] Eslami, SM Ali, et al. "Attend, infer, repeat: Fast scene understanding with generative models." Advances in Neural Information Processing Systems. 2016.
> > > [5] Greff, Klaus, et al. "Neural expectation maximization." Advances in Neural Information Processing Systems. 2017.
> > > [6] Greff, Klaus, et al. "Tagger: Deep unsupervised perceptual grouping." Advances in Neural Information Processing Systems. 2016.
> > > [7] van Steenkiste, Sjoerd, et al. "Relational neural expectation maximization: Unsupervised discovery of objects and their interactions." International Conference on Learning Representations. 2018.
> > > [8] Yang, Jianwei, et al. "LR-GAN: Layered recursive generative adversarial networks for image generation." International Conference on Learning Representations. 2017.
> > > [9] Zambaldi, Vinicius, et al. "Relational Deep Reinforcement Learning." arXiv preprint arXiv:1806.01830 (2018).

---

### Official Review · AnonReviewer2 · 2018-11-04
**An interesting method, but more experiments needed**

**Rating:** 5
**Confidence:** 5

**Review:**

[Overview]

In this paper, the authors proposed a compositional image generation methods that combines multiple objects and background into the final images. Unlike the counterpart which compose the images sequentially, the proposed method infer the relationships between multiple objects through a relational network before sending the hidden vectors to the generators. This way, the method can model the object-object interactions during the image generation. From the experimental results, the authors demonstrated that the proposed k-GAN can generate the images with comparable or slightly better FID compared with baseline GAN, and achieve plausible performance under the human study.

[Strengthes]

1. This paper proposed an interesting method for compositional image generation. Unlike the counterparts like LR-GAN, which generate foreground objects recurrently, this method proposed to derive the relationships between objects in parallel simultaneously. This kind of relational modeling has been seen in many other domains. It would be nice to see it can be applied to the compositional image generation domain.

2. The authors tried multiple synthesized datasets, including multi-MNIST and its variants, CLEVR. From the visualization, it is found that the proposed k-GAN can learn to disentangle different objects and the objects from the background. This indicates that the proposed model indeed capture the hidden structure of the images through relational modeling. The human study on these generated images further indicate that the generated images based on k-GAN is better than those generated by baseline GAN.

[Weaknesses]

1. The main contribution of this paper fall to the proposed method for modeling the relational structure for multiple objects in the images. In the appendix, the authors presented the results for the ablated version which does not consider the relationships. As the authors pointed out, these results are a bit counterintuitive and concluded that FID is not a good metrics for evaluating compositional generation. However, as far as I know, the compositional generation can achieve much better Inception scores on CIFAR-10 in LR-GAN paper (Yang et al). Combining the results on MNIST in LR-GAN paper, I would suspect that the datasets used in this paper are fairly simple and all methods can achieve good results without much efforts. It would be good to show some results on more complicated datasets, such as face images with background, or cifar-10. Also, the authors did not present the qualitative results for independent version of k-GAN. Meanwhile, they missed an ablated human study when the relational modeling is muted. I would like to see how the generated images without modeling relationships look like to humans.

2. Following the above comment, I think the datasets used in this paper is relatively simpler. In MM and CLEVR, the foreground objects are digits or simple cubes, spheres or cylinders. Also, the background is also simpler for these two datasets. Though CIFAR10+MM has a more complicated background, it is trivial for the model to distinguish the foregrounds from the backgrounds. Again, the authors should try more complicated datasets.

3. Though the proposed method can model the relationship between objects simultaneously, I doubt its ability to  really being able to disentangle the foregrounds from the backgrounds. Since the background and foregrounds are both whole images, which are then combined with an alpha blending, the model cannot discover the conceptually different properties for foreground and background that foregrounds are usually small than background and scattered at various locations. Actually, this has been partially demonstrated by Figure 4. In the last row, we can find one sphere is in the background image. I tend to think the proposed model performs similar to Kwak & Zhang's paper without a strong proof for the authors that the relational modeling plays an important role in the model.

4. It would be nice to perform more analysis on the trained k-GAN. Given the training set, like MM or CLEVR, I am wondering whether k-GAN can learn some reasonable relationship from the datasets. That is, whether it is smart enough to infer the right location for each objet by considering the others. This analysis can be performed, how much occlusions the generated images have compared with the real images. For example, on CLEVR, I noticed from the appendix that the generated CLEVR images base on k-GAN actually have some severe occlusions/overlaps.

[Summary]

In this paper, the authors proposed an interesting method for image generation compositionally. Instead of modeling the generation process recurrently, the authors proposed to model the relationships simultaneously in the hidden vector space. This way, the model can generate multiple foreground objects and backgrounds more flexibly. However, as pointed above, the paper missed some experiment, ablation study and analysis to demonstrate the relational modeling in the image generation. The author need to either try more complicated images or add deeper analysis on the recent experimental results.

---

> ### Author Response · Authors · 2018-11-16
> **Reply to Reviewer 2 [1 / 2]**
>
> Thank you for your consideration and feedback.
>
> The primary motivation of this work is to argue for object compositionality in deep generative models (and in particularly GANs), which originates from two key observations. First, real-world images are to a large degree compositional, and a generative model that is suitable equipped with a corresponding inductive bias should be better at capturing this distribution. Second, in disentangling information content corresponding to different objects at a representational level they may be recovered a posteriori unlike in unstructured models.
>
> In the following we will answer each of your comments.
>
> (1) Our conclusion regarding FID arises from the way the Inception network (that provides the embedding) was trained. In particular, by training on ImageNet for single-object classification it is unlikely that deep layers (eg. logits or final max-pool) provide high-level features that capture properties of multiple objects accurately. In particular, it suggests that FID is limited in accurately evaluating generated images containing multiple objects, even though it is accurate in evaluating generative models on ImageNet (or related single-object tasks like CIFAR10, etc.) as shown in [4]. Similarly, this does not preclude LR-GAN (or other compositional approaches) from using FID on ImageNet or CIFAR10.
>
> On the contrary, we compute FID on multi-object images using an inception network that was pre-trained on single-object images (ImageNet). We are interested in verifying that the generated images are faithful with respect to the training distribution in terms of the number of objects, their identities, etc. To the best of our knowledge FID has not been used in this way previously. LR-GAN [5] evaluates the Inception score only on MNIST-ONE and not on MNIST-TWO, although they conclude that it is unsuitable even in the single object case (see Appendix 6.3). Based on our own observations in using FID on multi-object datasets (as summarised in Figure 9) we argue that FID is unable to judge generated images based on specific properties relating to multiple objects (eg. their total number, etc.). The large differences that are observed in evaluating the subjective quality (human eval - Figure 6a) for models with similar FID provide additional evidence that this is the case.
>
> (2) In this work we are interested generating scenes as compositions of objects, and in particular in verifying that this information can be disentangled at a representation level. This requires evaluation on datasets for which a clear notion of “object” is available. Compared to prior work that has focused primarily on the representation learning part (eg. [1, 2, 3]), we focus on scaling these insights to more complex multi-object datasets.
>
> We would like to emphasize that relevant prior work has only focused on Multi-MNIST [1, 2, 3], Shapes [2, 3], and Textured MNIST [3]. In this work we consider several more complex datasets, including two relational version of Multi-MNIST (triplet, rgb), a variation on CIFAR10 that has RGB MNIST digits in the foreground, and high-resolution CLEVR images that contain many rendered geometric objects and require lighting and shadows to be modeled.
>
> Ideally we would be able to apply our approach to common segmentation datasets (eg. Pascal VOC, COCO) although in practice we find that these are still far out of reach for purely unsupervised approaches. Such datasets have been designed with access to ground-truth labels in mind and the large imbalance between the visual complexity of objects (i.e. intra-class variation) and the number of samples renders them unsuitable for our purpose. We consider CLEVR to be among the more complex multi-object datasets that are balanced in this way, and hence the feasibility of our approach on this dataset is an important step forward compared to prior work.

---

> > ### Author Response · Authors · 2018-11-16
> > **Reply to Reviewer 2 [2 / 2]**
> >
> > (3) The conceptual difference between foregrounds and backgrounds i.e. that foregrounds are usually smaller than backgrounds and scattered at various locations is encoded in the learned alpha channel. Each of the object (foreground) generators draws an object in the scene at a random location (based on the mask, and RGB values), requiring location and size to be encoded in the latent representation z_i. This is important as it allows an inference model to extract this information correspondingly.
> >
> > We would like to emphasize that the reported images in Figures 3 & 4 are representative of how the best performing models generate the scene. In other words, on all datasets we consistently find that the network generates the images as a composition of individual objects. Indeed on CLEVR we found cases in which a component generates more than one object, which is understood by the fact that the number of components is smaller than the number of objects in the scene (see also Figure 8 in Appendix A). We also find infrequent cases (primarily on CLEVR, although sometimes on CIFAR10) in which the background generator generates an additional object. This is understandable as there are no restrictions in our approach that prevent it from doing so. However, given that in almost all other cases the network generate images as compositions of objects and background it seems reasonable to conclude that these are due to optimization issues (as are common in GANs). After all, the compositional solution is clearly the superior choice, as is evident from our human evaluation compared to regular GAN.
> >
> > (4) We are unable to provide the experiment that you are proposing as unlike in [5] the MNIST digits considered in our approach do not appear at a fixed location. We did study other properties of the generated images as can be seen in Figure 6b, Figure 7, and Figure 8. In particular we find that k-GAN is better at generating 3 RGB digits (requiring the relation to be captured) compared to GAN as can be seen in Figure 6b and Figure 7. Secondly, the fact that increasing the number of components to 4 or 5 reduces the accuracy with which the correct number of digits is generated only marginally (eg. Figure 6b) provides additional evidence that the relational mechanism has learned about the correct number of objects.
> > Finally, the large difference in FID scores (Figure 9) in comparing k-GAN with and without a relational mechanism on rgb MM can only be explained by the relational mechanism correctly accounting for the different color of the digits.
> >
> > [1] Eslami, SM Ali, et al. "Attend, infer, repeat: Fast scene understanding with generative models." Advances in Neural Information Processing Systems. 2016.
> > [2] Greff, Klaus, et al. "Neural expectation maximization." Advances in Neural Information Processing Systems. 2017.
> > [3] Greff, Klaus, et al. "Tagger: Deep unsupervised perceptual grouping." Advances in Neural Information Processing Systems. 2016.
> > [4] Lucic, Mario, et al. "Are gans created equal? a large-scale study." arXiv preprint arXiv:1711.10337 (2017).
> > [5] Yang, Jianwei, et al. "LR-GAN: Layered recursive generative adversarial networks for image generation." International Conference on Learning Representations. 2017.

---

### Meta-Review · Area_Chair1 · 2018-12-14
**related literature and evaluations**

**Confidence:** 4
**Recommendation:** Reject

**Metareview:**

The paper proposes a generative model that generates one object at a time, and uses a relational network to encode cross-object relationships. Similar  object-centric generation and object-object relational network  is proposed in "sequential attend, infer, repeat" of Kosiorek et al. for video generation, which first appeared on arxiv on June 5th 2018 and was officially accepted in NIPS 2018 before the submission deadline for ICLR 2019. Moreover, several recent generative models have been proposed that consider object-centric biases,  which the current paper references  but does not compare against, e.g., 'attend, infer, repeat' of Eslami et al., or "DRAW: A Recurrent Neural Network For Image Generation" of Gregor et al. . The CLEVR dataset considered, though it contains real images, the intrinsic image complexity is low because it features a small number of objects against table background. As a result, the novelty of the proposed work may not be sufficient in light of recent literature, despite the fact that the paper presents a reasonable and interesting approach for image generation.